EMBO
Molecular Medicine

# Chromatin accessibility landscape of pediatric T-lymphoblastic leukemia and human T-cell precursors

Büşra Erarslan-Uysal[1,2,3,†] [iD], Joachim B Kunz[1,2,3,4,†], Tobias Rausch[3,5,6,†] [iD], Paulina Richter-Pechańska[1,2,3,†], Ianthe AEM van Belzen[5], Viktoras Frismantas[6], Beat Bornhauser[7], Diana Ordoñez-Rueada[8], Malte Paulsen[8], Vladimir Benes[6], Martin Stanulla[9], Martin Schrappe[10], Gunnar Cario[10], Gabriele Escherich[11], Kseniya Bakharevich[11], Renate Kirschner-Schwabe[12], Cornelia Eckert[12], Tsvetomir Loukanov[13], Matthias Gorenflo[14], Sebastian M Waszak[5], Jean-Pierre Bourquin[7], Martina U Muckenthaler[1,2,3], Jan O Korbel[3,5,*] [iD] & Andreas E Kulozik[1,2,3,**] [iD]

## Abstract

We aimed at identifying the developmental stage at which leukemic cells of pediatric T-ALLs are arrested and at defining leukemogenic mechanisms based on ATAC-Seq. Chromatin accessibility maps of seven developmental stages of human healthy T cells revealed progressive chromatin condensation during T-cell maturation. Developmental stages were distinguished by 2,823 signature chromatin regions with 95% accuracy. Open chromatin surrounding *SAE1* was identified to best distinguish thymic developmental stages suggesting a potential role of SUMOylation in T-cell development. Deconvolution using signature regions revealed that T-ALLs, including those with mature immunophenotypes, resemble the most immature populations, which was confirmed by TF-binding motif profiles. We integrated ATAC-Seq and RNA-Seq and found *DAB1*, a gene not related to leukemia previously, to be overexpressed, abnormally spliced and hyper-accessible in T-ALLs. *DAB1*-negative patients formed a distinct subgroup with particularly immature chromatin profiles and hyper-accessible binding sites for *SPI1* (*PU.1*), a TF crucial for normal T-cell maturation. In conclusion, our analyses of chromatin accessibility and TF-binding motifs showed that pediatric T-ALL cells are most similar to immature thymic precursors, indicating an early developmental arrest.

**Keywords** ATAC-Seq; chromatin accessibility; T-cell development; T-cell leukemia
**Subject Categories** Cancer; Chromatin, Transcription & Genomics

## Introduction

T-cell acute lymphoblastic leukemia (T-ALL) is considered to result from uncontrolled proliferation and developmental arrest at early stages of differentiation of normal T-cell progenitors. T-cell lineage commitment and differentiation is coordinated through an interplay between signals from the thymic microenvironment such as cytokines, chemokines and antigens and intrinsic transcription factors

1 Department of Pediatric Oncology, Hematology, and Immunology, University of Heidelberg, Heidelberg, Germany
2 Hopp Children's Cancer Center (KiTZ) Heidelberg, Heidelberg, Germany
3 Molecular Medicine Partnership Unit (MMPU), European Molecular Biology Laboratory (EMBL), Heidelberg, Germany
4 German Consortium for Translational Cancer Research (DKTK), Heidelberg, Germany
5 Genome Biology Unit, European Molecular Biology Laboratory (EMBL), Heidelberg, Germany
6 Genomics Core Facility, European Molecular Biology Laboratory (EMBL), Heidelberg, Germany
7 Division of Pediatric Oncology, University Children's Hospital, Zürich, Switzerland
8 Flow Cytometry Core Facility, European Molecular Biology Laboratory (EMBL), Heidelberg, Germany
9 Department of Pediatric Hematology and Oncology, Hannover Medical School, Hannover, Germany
10 Department of Pediatrics, University Hospital Schleswig-Holstein, Kiel, Germany
11 Clinic of Pediatric Hematology and Oncology, University Medical Center Hamburg-Eppendorf, Hamburg, Germany
12 Department of Pediatric Oncology/Hematology, Charité Universitätsmedizin Berlin, Berlin, Germany
13 Department of Cardiac Surgery, University of Heidelberg, Heidelberg, Germany
14 Department of Pediatric Cardiology and Congenital Heart Diseases, University of Heidelberg, Heidelberg, Germany
*Corresponding author. Tel: +49 6221 3878822; E-mail: jan.korbel@embl.org
**Corresponding author. Tel: +49 6221 56 4500, +49 6221 56 4555; E-mail: andreas.kulozik@med.uni-heidelberg.de
†These authors contributed equally to this work

which regulate and orchestrate the process (Yui & Rothenberg, 2014). Developing T cells are conventionally classified by the expression of surface antigens (Bene *et al*, 1995). T-cell precursors that immigrated from the bone marrow undergo a Notch signaling dependent commitment to early double-negative stages (CD4$^-$ and CD8$^-$; DN1–DN2) (Osborne & Miele, 1999). Thymocytes failing to produce a functional pre-TCR by V(D)J rearrangement of T-cell receptors are eliminated by apoptosis (ß-selection), while the remaining cells develop beyond the DN3-ISP stage. Following positive and negative selection of double-positive (DP: CD4$^+$ and CD8$^+$) and single-positive stages (SP: CD4$^+$ or CD8$^+$), only those thymocytes with a T-cell receptor capable of binding MHC molecules, but without high affinity for self-antigens (Raulet *et al*, 1985; Pardoll *et al*, 1987), survive and differentiate into either single-positive CD4$^+$ T helper cells or CD8$^+$ T cytotoxic cells (Zerrahn *et al*, 1997).

T-cell leukemias are historically subclassified in analogy to T-cell maturation by the expression of the surface markers into four immunophenotypes: pre-, pro-, cortical- and mature-T-ALLs, respectively (Bene *et al*, 1995). Additionally, T-ALLs arrested at the early thymic progenitor (ETP) stage (Hosokawa & Rothenberg, 2018) have been recognized as a distinct subgroup (Coustan-Smith *et al*, 2009; Inukai *et al*, 2012). A hallmark of T-ALL is the rearrangement and activation of the oncogenic transcription factors *TAL1, LMO2, TLX3*, HOXA and NKX2 whose promoters are often placed under the control of strong T-cell-specific enhancers. In addition, the genomic landscape of T-ALL often includes "type B" mutations such as activating *NOTCH1*-pathway mutations, deletions of *CDKN2A*, activation of the *IL7R*/JAK-STAT pathway, and loss of the *PTEN* tumor suppressor gene (Van Vlierberghe *et al*, 2008; Girardi *et al*, 2017).

While cytogenetics and cell surface marker staining are used to characterize the putative cell of origin of T-ALL, epigenomic analyses have recently evolved as powerful methods to define the biology of tumor cells and their relationship to the normal precursors (Corces *et al*, 2016; Beekman *et al*, 2018). To identify the maturation stages closest to T-ALLs and thus where T-ALL cells are likely arrested, we employed the Assay for Transposase Accessible Chromatin Sequencing (ATAC-Seq) and analyzed the chromatin landscape for different developmental stages of sorted healthy thymic progenitors obtained from otherwise healthy children undergoing heart surgery. We generated a comprehensive map of stage-specific regions of chromatin accessibility in the course of human T-cell development and predicted transcription factor (TF)-binding motifs in the footprints of ATAC peaks enriched in each developmental stage. The signature served to identify T-cell maturation stages that most closely resembled the ATAC landscape of pediatric T-ALL. Finally, we integrated RNA-Seq with ATAC-Seq data and identified overexpressed genes residing in highly accessible chromatin and playing previously unknown roles in the biology of pediatric T-ALL.

# Results

## The genome-wide landscape of chromatin accessibility undergoes gradual changes during T-cell development

Using ATAC sequencing, we generated chromatin accessibility maps of seven populations of sorted healthy T-cell precursors (DN2, DN3,

ISP, DPCD3$^-$, DPCD3$^+$, SPCD4$^+$, and SPCD8$^+$) obtained from thymi of six otherwise healthy children undergoing heart surgery (Fig 1A). Of the 68,415 ATAC-Seq peaks (open chromatin regions; OCRs) (Dataset EV1) identified in the combined analysis of the six donors, the majority (85.2%; $n = 58,294$) fell into distal (non-TSS; outside $\pm 1$ kb window of a TSS) regions, while 14.8% ($n = 10,121$) fell into regions of transcription start sites (TSS; $\pm 1$ kb of a TSS). As previously shown, distal regulatory elements more accurately classify different cell populations (Heinz *et al*, 2010; Corces *et al*, 2016) than TSS regions and were therefore used in downstream analyses in our study unless stated otherwise.

Chromatin accessibility as measured by the number of OCRs decreased progressively with maturation and reached a minimum in the DP and SP stages (Fig 1B, *P*-value = 0.035; Kruskal–Wallis Test). These data indicate that the chromatin of developing T cells becomes increasingly condensed with maturation, which is consistent with previous observations that chromatin in undifferentiated embryonic stem cells is globally decondensed in comparison to differentiated cells (Gaspar-Maia *et al*, 2011; Ugarte *et al*, 2015). Consequently, the number of accessible TF-binding motifs identified in OCR footprints of seven developmental stages decreased significantly during T-cell development (Appendix Fig S1A, *P*-value = 0.0028; Kruskal–Wallis Test). Moreover, we have categorized 68,415 OCRs into four patterns (increasing, decreasing, fluctuating, and steady) based on the changes in accessibility patterns and found that the majority of OCRs (59%) remain steady during thymocyte maturation (Fig 1C). Steady peaks tended to have less accessibility (mean peak count: 20), indicating that less open chromatin regions in early development tend to remain closed as T cells develop. We found that 29% of OCRs became less accessible, whereas only 0.3% became more accessible demonstrating that chromatin organization in developing thymocytes is characterized by closing/condensing those regions that are highly accessible in the immature precursors (Fig 1C).

Unsupervised learning by principal component analysis (PCA) of all distal OCRs grouped the sorted populations according to their developmental stage (Fig 1D). Arrangement of the groups on the PCA plot followed the hierarchy of their development and exhibited overlapping profiles thus indicating a continuous process of chromatin remodeling during T-cell maturation (Fig 1D). Quality control of the libraries revealed that potential batch effects do not detectably influence the data quality and do not drive the clustering of samples in the PCA (Dataset EV2). Moreover, the projection of bulk thymus in the PCA space of sorted T cells is concordant with FACS results showing that the majority of the cells in thymus belong to DPCD3$^+$ and DPCD3$^-$ populations (mean DP 70%; Appendix Fig S2).

A comparison with previously published methylation/acetylation datasets for the T-ALL cell line DND-41 (Knoechel *et al*, 2014), in which we computed expected values based on randomly shuffled ATAC-Seq peaks, shows a high degree of overlap between the OCRs identified in purified subpopulations and the active promoters and enhancers detected in ChIP-Seq datasets (Fig EV1).

## Developmental stage-specific conformation of chromatin reveals key regulatory regions of T-cell development

To define a set of OCRs differentially accessible across developmental stages, we performed pairwise contrasts of the different sorted T-cell precursors using DESeq2 (Love *et al*, 2014). Out of 58,294

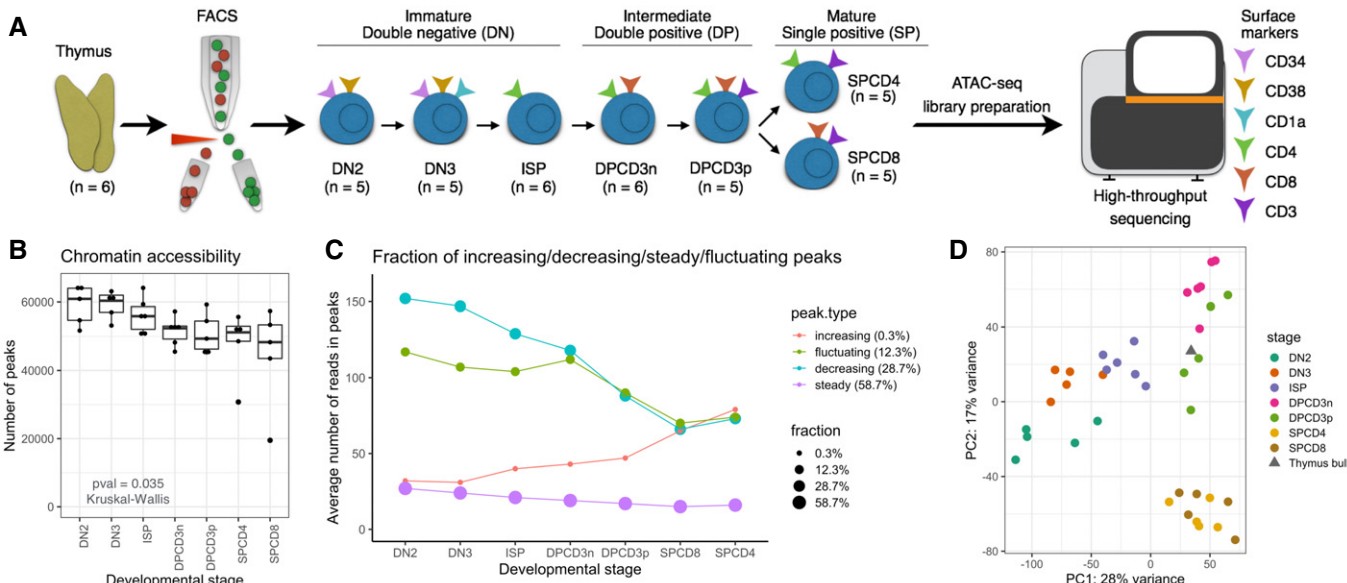

**Figure 1. Chromatin accessibility by ATAC-Seq differentiates maturation stages of healthy T-cell precursors.**

A   Thymi collected from healthy donors (*n* = 6) were sorted by FACS to obtain seven populations of double-negative (DN2, DN3, and ISP), double-positive (DPCD3⁻ and DPCD3⁺), and single-positive (SPCD4⁺ and SPCD8⁺) stages. Sorted populations were subjected to the ATAC sequencing.

B   Chromatin accessibility as measured by the number of all (TSS and distal) ATAC-Seq peaks (read count ≥ 10) detected in DN2 (*n* = 5), DN3 (*n* = 5), ISP (*n* = 5), DPCD3⁻ (*n* = 6), DPCD3⁺ (*n* = 6), SPCD4⁺ (*n* = 5), SPCD8⁺ (*n* = 5) populations. Horizontal lines of the box plot indicate the median, lower, and upper limits of each box correspond to the first and third quartiles (the 25th and 75th percentiles) and the lower and upper whiskers extend from min to max; *P*-value = 0.035 (Kruskal–Wallis test).

C   Fraction of OCRs having increasing, decreasing, steady, and fluctuating accessibility pattern during T-cell development (*x*-axis). Average number of normalized reads in OCRs (*y*-axis).

D   Unsupervised learning by PCA of all distal peaks. Color represents developmental stage. The gray triangle indicates the projection of the bulk thymus sample on the PCA space of sorted populations.

distal OCRs, we identified a signature set of 2,823 OCRs that distinguish T-cell precursors (Dataset EV3). A heatmap of all signature regions clustered seven T-cell populations in the order of their maturation highlighting the continuous changes in chromatin accessibility during T-cell maturation (Fig 2A). Gradual changes in consecutive developmental stages resulted in poor separation particularly between DN3 and ISP, and double-positive stages CD3⁻ and CD3⁺ (Fig 2A) resulting in a suboptimal prediction accuracy in leave-one-out cross-validations (Appendix Fig S3A and B) using the CIBERSORT (Newman *et al*, 2015) deconvolution algorithm to predict cell types. Moreover, a time course experiment showed rapid internalization of surface CD3 when DPCD3⁺ cells were incubated on ice, indicating that the expression of CD3 on the surface DP cells is not related to major functional differences (Appendix Fig S4). Therefore, we merged ATAC-Seq datasets of populations that showed most commonalities and used the following five groups for all downstream differential analyses: DN2, DN3 & ISP, DPCD3⁻ & DPCD3⁺, SPCD4, and SPCD8 (Fig EV2; see Materials and Methods—Generation of signature matrix for more details).

Using the rotations of principal components 1 and 2 (PC1 & PC2), we quantified the importance of signature OCRs and found that a peak on the SUMO1 Activating Enzyme Subunit 1 (*SAE1*; Fig 2B), a crucial element of the SUMO modification system, contributes most to PC1 differentiating developmental groups thus indicating that SUMOylation may play an important role in T-cell development. Further, SUMO-specific peptidase 3, 5, and 7 (*SENP3,*

*5, and 7*), E3 SUMO-protein transferase ERG2, *BCL11A*, which colocalize with *SUMO1* and *SENP2*, and two other genes contained in the SUMOylation pathway (*TPR* and *NUP214*) were included in the signature. Two out of the top five signature OCRs with the highest PC1 contribution are identified near the T-cell receptor gamma (TRG) locus (Dataset EV3) which is rearranged resulting in the gamma-delta T-cell lineages to branch off from double-negative populations thus separating this profile from that of the alpha-beta T-cell lineage (Dataset EV3). Moreover, in the set of signature OCRs we recapitulated differentially accessible OCRs surrounding other well-known stage-specific genes such as *CD8A* (Ellmeier *et al*, 1999) and *RAG1/2* (Rothenberg *et al*, 2008) (Fig 2B, Dataset EV3).

Functional enrichment analysis (FEA) by GREAT (McLean *et al*, 2010) showed that signature OCRs are enriched in the proximity of genes related to T-cell development and the function of the immune system (Dataset EV4). Signature OCRs assigned to the double-positive group (790; Appendix Fig S5) were enriched in terms associated with hemopoiesis and leukocyte differentiation and those assigned to the single-positive group (1,169; SPCD4⁺ and SPCD8⁺, Appendix Fig S5) were enriched in terms related to the regulation of the immune system (Fig 2C and Dataset EV4). However, signature peaks (864; Appendix Fig S5) in the double-negative group (DN2 and DN3&ISP) had only one exclusively enriched GO-term, namely positive regulation of leukocyte migration, suggesting that OCRs regulate more general functions as long as cells retain a multi-lineage capacity (Fig 2C and Dataset EV4).

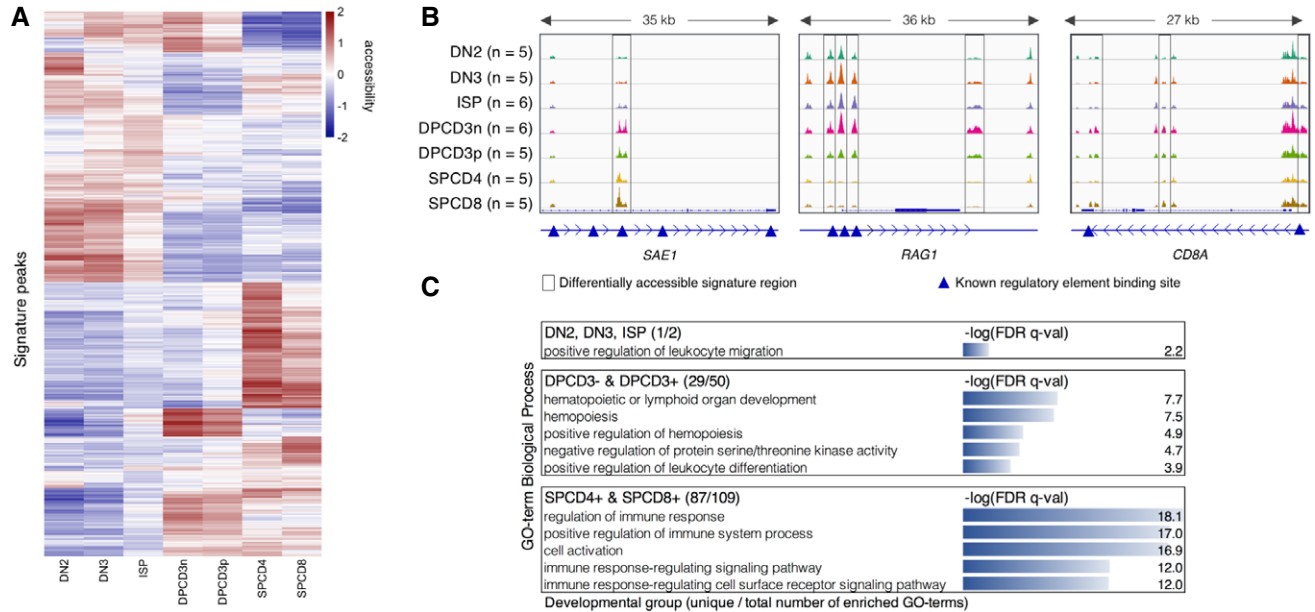

**Figure 2. Signature open chromatin regions distinguish T-cell developmental stages.**

A   Unsupervised hierarchical clustering based on the variance stabilizing transformation (VST) normalized read counts (each row is also normalized by the row mean) of the 2,021 signature open chromatin regions of 7 stages (see Fig EV2 for five stages).

B   ATAC-Seq genome tracks showing three signature peaks located near the genes *SAE1*, *RAG1*, and *CD8A*. Framed regions indicate cell-type-specific signature ATAC-Seq peaks. Blue triangles indicate known promoter or enhancer regions, which are annotated in the GeneHancer (GH) Regulatory Elements database (Fishilevich *et al*, 2017).

C   Functional enrichment analysis of the signature OCRs allocated to three developmental groups by GREAT. Top five unique GO-terms which have the lowest FDR q-val are shown. In parenthesis: number of uniquely enriched GO-terms / total number of enriched GO-terms.

## Differential motif enrichment analysis reveals key transcription factors regulating T-cell development

We annotated ATAC-Seq peaks with TF-binding sites using Alfred (Rausch *et al*, 2019) and the JASPAR motif database (Khan *et al*, 2018). To enhance specificity of motif predictions, we further intersected candidate motif-binding sites with TF footprinting predictions from HINT-ATAC (Li *et al*, 2019) (Fig 3A). High motif counts (> 1,000) for well-known T-cell-related TFs such as TCF3, RUNX1, ETS1, and HES1 (Rothenberg *et al*, 2008) confirmed the accuracy of our computational predictions in the footprints within OCRs.

Of altogether 354 analyzed TF-binding motifs, 77 (21%) were specifically accessible in one developmental stage compared to the other stages ($P_{adj} < 0.05$; Fisher's exact test; Dataset EV5). The majority (44) was enriched in the most immature stage DN2 (Appendix Fig S6A), whereas in the other populations we identified only 5–11 stage-specific TF-binding motifs (Appendix Fig S6A). Moreover, the majority (49/84; 58%) of TF-binding motifs specifically depleted in OCRs of one developmental stage were identified in the DP populations (Appendix Fig S6B). In order to assess if the enrichment of TF-binding motifs in OCRs can be used to predict the differential activity of TFs during T-cell development, we focused on the top 50 TFs with the highest standard deviation of odds ratios between the developmental stages. We first confirmed the hyperaccessibility of TF motifs whose activities are known to be restricted to certain developmental stages (Dataset EV5). These, marked by green squares (18) in Fig 3B, include SPI1 (PU.1, highest

discriminating capacity; OR = 4.1 in DN2), Bach2, IRF1, and IRF2 (Ungerback *et al*, 2018) (Simon *et al*, 1997; Anderson *et al*, 2002a). Further, these data reveal development-specific activity of 25 TFs that have previously been linked to T-cell development but have so far not been characterized for their differential activity during T-cell maturation. These TFs are listed in Dataset EV5 and include MAF:: NFE2, Bach1::Mafk, ETS-related family TFs ELF1/4, ELK1/3, and ETV3 (marked by pink squares in Fig 3B). Finally, we identify seven additional TFs (blue squares in Fig 3B) that so far have not been linked to T-cell development. Two of these are enriched in a stage-specific manner (ONECUT1 and PAX3), whereas the remaining five are enriched either in early (DN2, DN3&ISP) populations (ZBTB18) or in intermediate and mature (DP, SPCD4, and SPCD8) populations (ZBTB7A, USF2, ATF7, and HINFP).

Furthermore, we have performed a global TF motif enrichment analysis on the ATAC-Seq peaks alone and compared the results of this analysis to enrichment analysis of motifs occurring only within footprints identified in ATAC peaks. Global analysis yielded a higher number of TFs with no prior known relevance to T-cell biology than intersecting peak calls with footprints (20 vs. 7 TFs; Fig EV3), indicating that footprinting analysis adds substantial specificity toward identifying T-cell-related factors.

We have next investigated the hierarchical trees defined by JASPAR (JASPAR clusters), which are based on binding motif similarities. This analysis showed that our top 50 TFs belong to 20 distinct clusters (Dataset EV5). We found that 7/11 (64%) TFs (including IRF1–9 and Stat1–2) that are known to be highly relevant to

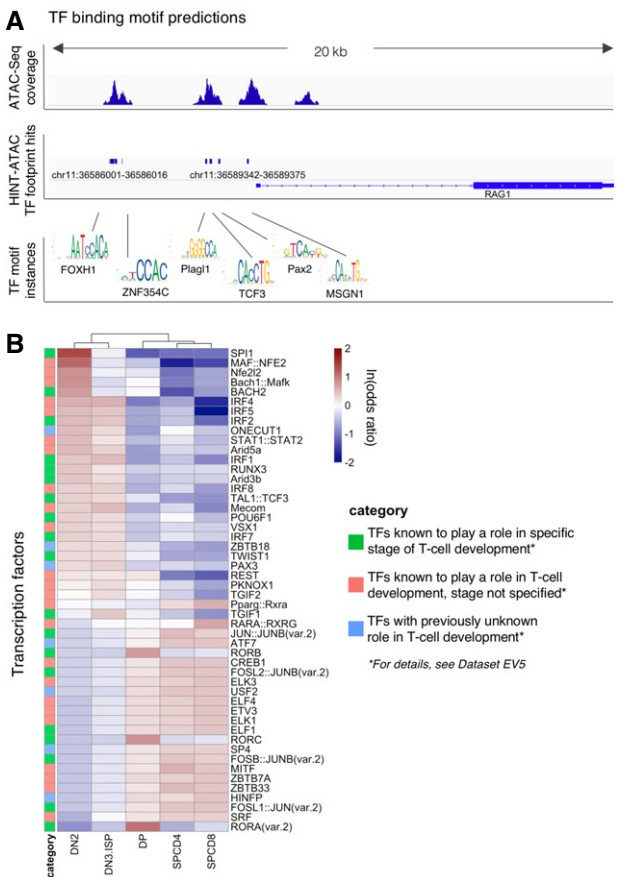

**Figure 3. Open chromatin regions contain maturation-specific transcription factor-binding motifs.**

A The rationale of footprinting analysis as illustrated by actual ATAC-Seq peaks, HINT-ATAC footprint hits, and example TF motif instances near *RAG1* gene in DN2 population of thymus donor 6.

B Heatmap of ln(odds ratio) of the 50 TFs showing the most differential abundance of binding motifs in the open chromatin regions. TFs with an SD of the motif counts < 10 are excluded.

T-cell development belong to the most represented cluster39 and are among our top candidates. When inspecting the clusters of seven TFs with no prior known relevance to T-cell biology, we noted that the binding motifs of two of these (HINF and SP4) do not show similarities to other top candidates (Dataset EV5). HINFP is the only TF in cluster76 and SP4 is the only TF among our top 50 belonging to cluster34, which includes a total of 29 TFs, suggesting a previously unrecognized role of HINFP and SP4 in T-cell differentiation. By contrast, the binding motifs of five of these seven TFs show similarity with the binding motifs of other members in the list of top 50 (Dataset EV5), and thus, these TFs may merely confirm the role of previously known TFs.

**Immature thymic T-cell precursors share open chromatin regions with T-ALL**

A map of chromatin accessibility and TF-binding motifs of the five developing healthy T-cell populations (DN2, DN3&ISP, DPCD3$^-$&DPCD3$^+$, SPCD4$^+$, and SPCD8$^+$) served as a basis for the deconvolution of 19 pediatric T-ALLs reflecting different immunophenotypes (pre-, pro-, cortical-, and mature-T-ALL; Dataset EV6). Seven out of 29 PDX models corresponding to 4/19 patients with T-ALL (Fig 4A) were previously shown to recapitulate the chromatin accessibility landscapes of primary leukemia samples and, in comparison to primary patient samples, to yield technically comparable material resulting in higher TSS enrichment scores of the ATAC library (P-value = 0.03; Wilcoxon test; (Richter-Pechanska et al, 2018)). The projection of the leukemia samples into the PCA space of 2,823 T-cell signature OCRs, which distinguished sorted healthy T-cell populations, positioned them in the vicinity of the early DN2 and DN3&ISP developmental stages (Fig 4B).

We next quantified the contribution of each of the developmental stages using the deconvolution algorithm CIBERSORT (Newman et al, 2015) trained on the 2,823 signature OCRs. Of the 19 leukemias, 16 had a contribution of at least 40% of one of the early developmental groups DN2 or DN3&ISP (Fig 4C). The average contribution of these early immature stages (DN2, and DN3&ISP) was 64% (DN2 = 36%, DN3&ISP = 28%), while the contribution of double positives and single positives was much lower (DP = 9%, SPCD4 = 13%, SPCD8 = 13%). Signature OCRs specific to the most immature T cells were particularly dominant (> 60%) in 4/19 T-ALL samples (DN2 > 60%; patients P3, P4, and P27, and DN3&ISP > 60%; P11). Notably, even the T-ALLs classified as mature by immunophenotyping exhibited a substantial contribution of the early DN2 chromatin accessibility landscape (P8: 52% and P27: 78%) indicating that even the subtype of T-ALL that is characterized by a mature immunophenotype resembles an immature thymic precursor (Fig 4C).

Chromatin accessibility, as measured by the number of ATAC-Seq peaks, decreased with the maturation of T cells (Fig 4D). Although T-ALLs (average peak count: 68,208) resembled most the early DN2 populations (average: 59,120), the number of peaks in T-ALLs was significantly higher than those of sorted T-cell populations (Fig 4D, P-value = 4.4e-08; Kruskal–Wallis test). Similarly, in the course of T-cell maturation we observed a decrease in the number of TF motifs that are located in OCR footprints. By contrast, the number of TF motifs counted in T-ALLs was significantly higher than those of sorted normal human T-cell populations (Appendix Fig S1B, P-value = 2.6e-05; Kruskal–Wallis Test).

Transcription factor-binding motifs analyzed in healthy T cells clustered these cells in two major developmental groups: early (DN2 and DN3&ISP) and intermediate together with mature T cells (DP, SPCD4$^+$ and SPCD8$^+$; Fig 3B). Out of the total of 354 analyzed TF-binding sites, 132 characterized early developmental stages ($P_{adj} < 0.05$ and ln[odds ratio] > 0) and 115 characterized intermediate and mature developmental stages (Dataset EV5). Out of the 132 early-specific TF-binding sites, such as motifs for Bach1/2, Maf (k), and Nfe2l2, 109 (83%) were significantly enriched in leukemias, whereas of the 115 intermediate- and mature-specific TFs only 12 (10%) were significantly enriched in T-ALLs (top 50 shown in Fig 4E). Binding motifs for TFs which characterized normal DP and SP stages, such as ZBTB7a, RORA/C, and CREB1, tended to be depleted in T-cell leukemias (Fig 4E).

**The chromatin of T-ALLs is substantially more accessible than that of normal human immature T-cells**

After identifying features of chromatin architecture shared between leukemias and healthy T cells in different stages of their

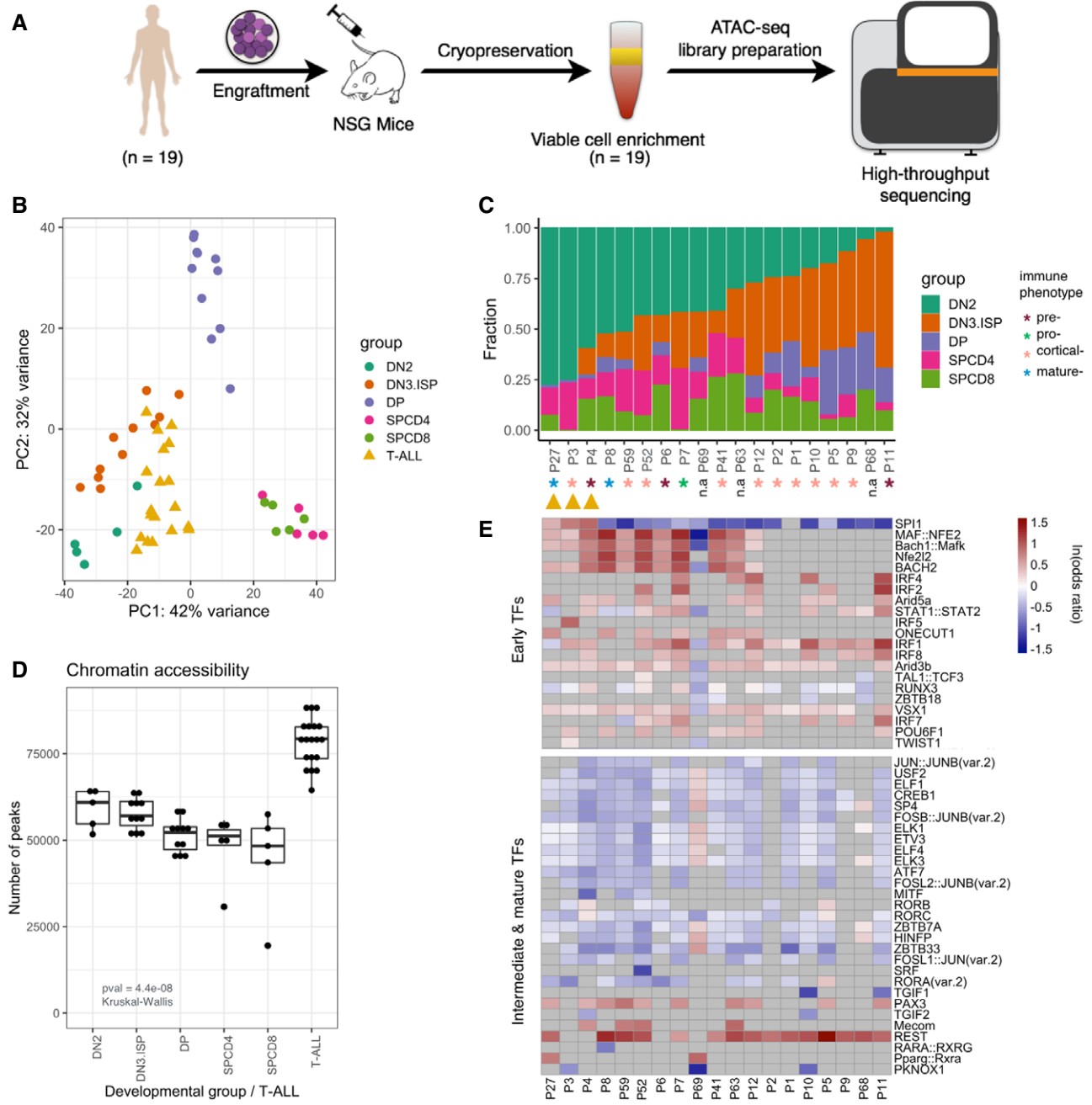

**Figure 4. The chromatin accessibility landscape of T-ALLs is most similar to the immature stages of normal human T-cell development.**

A  T-ALL cells of *n* = 19 patients were engrafted into mice, cryopreserved after harvesting and enriched for viable cells after defrosting. Then, they were subjected to the ATAC sequencing. Patients P7, P8, P10, P59 were published previously (Richter-Pechanska *et al*, 2018)

B  Projection of 19 T-ALLs in the PCA space of 2,823 signature OCRs discriminating T-cell developmental stages.

C  Deconvolution analysis by CIBERSORT trained on the 2,823 signature OCRs. Stacked columns represent the contribution of each developmental stage (indicated by color) to each leukemia patient (*x*-axis).

D  Chromatin accessibility as assessed by the number of ATAC peaks identified in the sorted populations of normal thymic T-cell precursors and in T-ALLs, respectively (*n* = 10/19 patients have two biological replicates, *n* = 9/19 patients have one replicate). Horizontal lines of the box plot indicate the median, lower, and upper limits of each box correspond to the first and third quartiles (the 25th and 75th percentiles) and the lower and upper whiskers extend from min to max; *P*-value = 4.4e−08 (Kruskal–Wallis test).

E  Heatmap representing the enrichment and depletion of binding motifs of top 50 TFs in OCRs of T-ALLs in comparison to healthy T-cell precursors. Patients on the *x*-axis; same order as in panel (C). Gray cells represent TFs having non-significant enrichment or depletion.

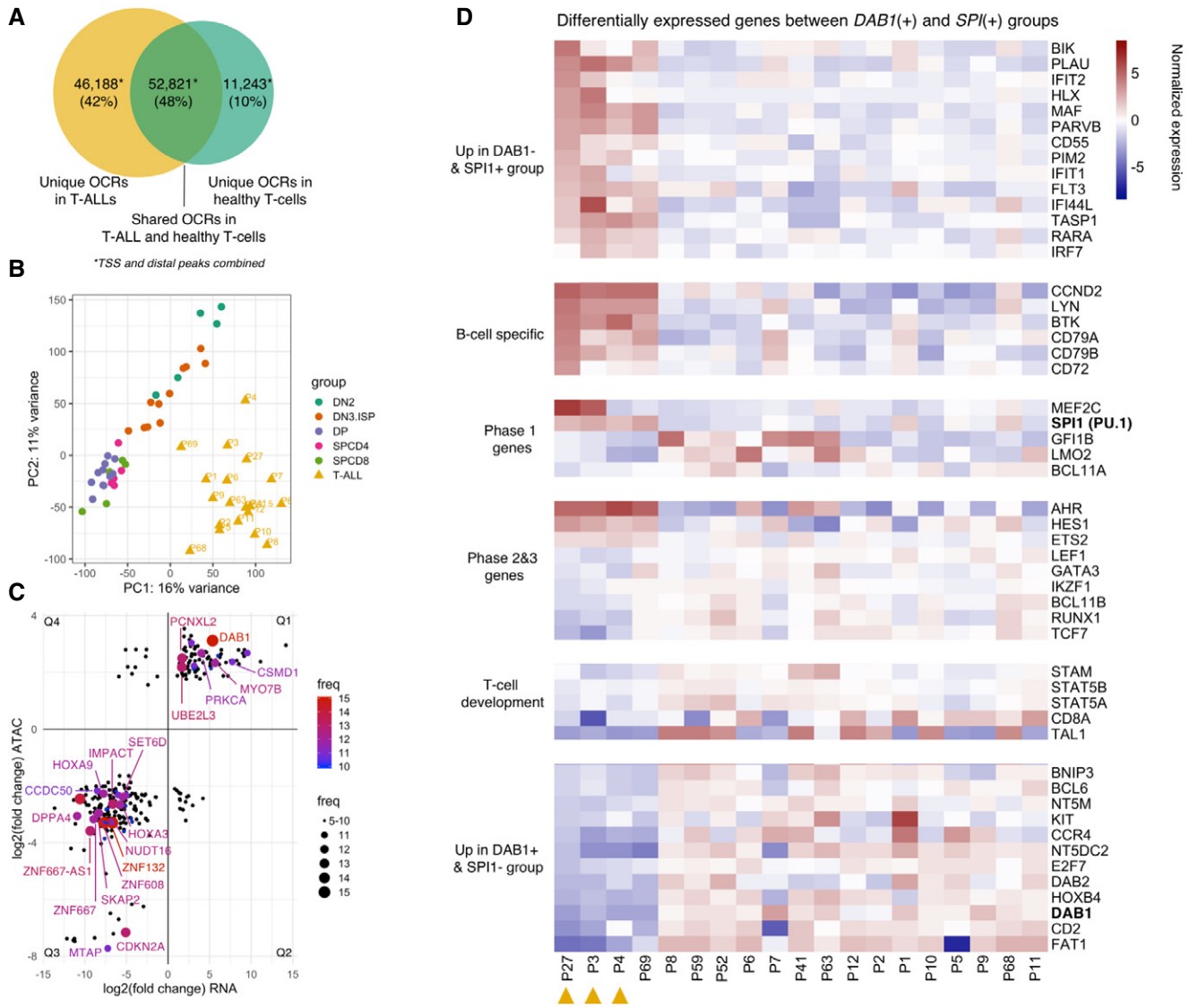

**Figure 5. The integration of differential chromatin accessibility and expression analyses reveals recurrently dysregulated genes.**

A Number of unique and shared OCRs (TSS and distal) in T-ALLs and in healthy T-cell precursors.

B Unsupervised learning by PCA of all TSS and distal peaks collected in the combined analysis of healthy and leukemic T cells.

C Log₂ fold changes of 292 genes which are differentially expressed (x-axis, RNA-Seq: non-sorted bulk thymi vs. T-ALLs) and differentially accessible (y-axis, ATAC-Seq: sorted T-cell populations vs. T-ALLs) in T-ALLs in comparison to healthy T-cell precursors. Dot size and color indicate the frequency (number of patients having dysregulation of the gene). Genes with ≥ 5/19 frequency are shown on the plot. Protein coding genes with ≥ 12/19 frequency are labeled. For plotting, the average LFC values of ATAC and RNA are used per gene per quartile. Q1–4 stands for the quadrant 1–4.

D Heatmap of VST normalized read counts (each row is also normalized by the row mean) of differentially expressed genes (P-value < 0.1; DESeq2) in three patients in comparison to remaining 16. Orange triangles indicate three patients with no *DAB1* but high *SPI1* expression and motif counts.

development, we were interested in genuine chromatin hallmarks of T-ALL. Of the 110,252 OCRs (non-TSS: 98,749, TSS: 11,503; Dataset EV7) identified in the combined analysis of sorted thymic T-cell populations and 19 T-ALLs, 52,821 were shared between healthy T-cell precursors and T-ALLs. By contrast, 46,188 OCRs (non-TSS: 44,323, TSS: 1,865) were accessible only in T-ALLs and 11,243 OCRs (non-TSS: 10,852, TSS: 391) were accessible only in healthy T-cell precursors (Fig 5A). The T-ALL chromatin landscape was thus substantially more accessible than that of even the most immature of the sorted normal T-cell precursor population indicating a

particularly immature profile in the leukemias. The majority of T-ALL-specific OCRs (44,323/46,188; 96%) were mapped to non-TSS OCRs suggesting that leukemia-specific OCRs are enriched for putative gene regulatory regions with an at least three-fold increase in accessible motifs for TFs with a key role in T-ALL such as REST and MAF:NFE2 (Dataset EV8).

Moreover, unsupervised learning by PCA using all OCRs resulted in a clear separation of T-ALL and healthy T cells demonstrating that the chromatin profiles of T-ALLs diverge fundamentally from those of healthy T-cell precursors in the thymus (Fig 5B). Quality

control of the libraries revealed that the higher number of OCRs in T-ALLs and the separation of T-ALLs and sorted T cells are not driven by potential batch effects or other technical parameters detailed in Dataset EV2.

### T-ALLs form two subgroups characterized by overexpression of *DAB1* or *SPI1*

We next integrated ATAC-Seq and RNA-Seq data thus focusing on chromatin regions and differentially expressed genes with a more likely biological relevance. Because of the high interpatient heterogeneity (Fig 5B), differential analyses by DESeq2 were performed separately for each of the patients, which enabled us to identify recurrent events (Dataset EV9). When compared to healthy T-cell precursors, we identified a total of 292 genes to be both differentially expressed and contained in significantly differentially accessible OCRs in at least five T-ALLs (Fig 5C). Of these peak/gene combinations, 102 were more accessible and overexpressed (quadrant—Q1), whereas 191 were less accessible with low RNA expression levels (Q3; Fig 5C).

The validity of this analysis was confirmed by the finding that the *CDKN2A* gene was identified to be not accessible/hypo-accessible and not to be expressed/under-expressed in the 16 patients with a homozygous *CDKN2A* deletion and one patient with a heterozygous *CDKN2A* deletion, respectively (data not available for two patients; Dataset EV10). *CDKN2A* was found to be among the most recurrently differentially accessible and expressed T-ALL-specific regions and contributes most to the discrimination between leukemias and healthy T cells (PC1 loading = 0.000597; Appendix Fig S7 and Dataset EV11).

The most recurrent differentially accessible OCR in T-ALLs (15/19; 79%) was assigned to the regulatory region of *DAB1* (PC1 loading = 0.000454, Appendix Fig S7 and Dataset EV11). The expression of *DAB1* was 41-times higher in the T-ALLs than in bulk thymus (fold change min: 14, median: 37, max: 218; DESeq2). We further validated *DAB1* expression by qPCR in four T-ALL cell lines, in four samples obtained from T-ALL patients, and in healthy bulk thymus (PCC = 0.97; Fig EV4), which confirmed the strong overexpression of this gene in the T-ALLs. We have also analyzed publicly available microarray expression profiling data of sorted human thymocyte populations (GEO Accession No.: GSE33470) and showed that *DAB1* expression is exceedingly low and similar to the expression of hemoglobin subunit alpha 1 (*HBA1*), which is known not to be expressed in T cells in all subpopulations (Appendix Fig S8). These data indicate that RNA-Seq of bulk thymi is representative for the expression of *DAB1* in all subpopulations.

Moreover, publicly available datasets generated with Affymetrix microarrays (U133) in larger patient cohorts demonstrated higher expression signal of *DAB1* in leukemias, particularly in T-ALLs (Appendix Fig S9; R2: Genomics Analysis and Visualization Platform). The role of *DAB1* in T-ALL is unexpected, because the biological function of this gene has so far been best documented in neurodevelopment (Bock & May, 2016; Lee & D'Arcangelo, 2016). Further, we discovered that T-ALLs express a novel transcript of DAB1 that is not annotated in the GTEx database but expressed in T-ALLs (Fig EV5). Expression of exons 10–15, which are not normally co-expressed, was validated by RT–PCRs and Sanger sequencing in two patients (P7 and P9) and three T-ALL cell lines

(Fig EV5). The integration of ATAC-Seq and RNA-Seq data combined with an exon usage analysis thus revealed that the leukemias overexpress a previously unannotated isoform of DAB1 including additional exons. This discovery indicates that the combination of methodologies used here is sensitive to uncover leukemia-specific abnormalities that were not called by previous RNA-Seq analyses alone. Future work will be required to investigate whether this previously unrecognized transcript of DAB1 affects the viability or proliferation of leukemia cells in T-ALL.

Four patients who did not express *DAB1* (P3, P4, P27, and P69) formed a distinct subgroup (Appendix Fig S10) characterized by *SPI1* overexpression, a particularly high contribution (> 60%) of the early developmental stage DN2 (Fig 4C) and a high accessibility of the binding motif of the TF SPI1 (PU.1) in three patients (Fig 4E). As expected, normalized read counts generated by RNA-Seq analysis showed a high correlation with counts for the SPI1-binding motifs in OCRs (Spearman's rank correlation = 0.72, *P*-value = 0.00048; Appendix Fig S11).

Differential analysis of gene expression and chromatin accessibility showed that *SPI1* expressing patients are characterized by an overexpression/hyper-accessibility of genes which determine the very early phase 1 of T-cell development (Seki *et al*, 2017). These genes included *MEF2C*, *MEIS1*, and *HOXA9* (Fig 5D, $P_{adj} < 0.05$; DESeq2) and genes that are characteristic for the B-cell lineage precursors such as *LYN*, *BTK*, cyclin D2, *CD79A*, *CD79B*, and *CD72* (Fig 5D, $P_{adj} < 0.05$; DESeq2). These findings indicate a particularly early cell type of origin of these T-ALLs with shared features of the B-cell lineage. By contrast, *DAB1* overexpressing and hyper-accessible T-ALLs tend to overexpress genes of the later phase 2 and 3 of T-cell development (Seki *et al*, 2017) such as *TCF7*, *LEF1*, and *RUNX1* (Fig 5D, $P_{adj} < 0.05$; DESeq2). In sum, these data implicate a previously unrecognized and mutually exclusive biology of *DAB1* and *SPI1*, and place the T-cell developmental stage, pediatric T-ALLs likely originate, at different albeit consistently early levels of the hierarchy of the developmental pathway of maturing T cells.

## Discussion

It has recently emerged that the differentiation and maturation of cells is paralleled by robust epigenomic changes reflected by DNA methylation and chromatin accessibility (Corces *et al*, 2016; Beekman *et al*, 2018; Yoshida *et al*, 2019). As a result, epigenetic analyses carry the potential to define the lineage and identity of cells more accurately than conventional methods (Corces *et al*, 2016; Rendeiro *et al*, 2016). Previous epigenomic analyses of pediatric T-ALL have demonstrated that T-ALL patients with a CpG island methylator-negative phenotype (CIMP-) reflecting generally low DNA methylation levels have a significantly worse clinical outcome in comparison to CIMP[+] patients (Borssen *et al*, 2016). Considering that medically most relevant features of pediatric T-ALLs such as treatment resistance have not been consistently possible to define by genetic and genomic analyses (Van Vlierberghe & Ferrando, 2012), we reasoned that epigenetic profiling of these leukemias and defining the thymic maturation stage leukemia cells likely originate may unveil previously unrecognized mechanisms of leukemogenesis in this challenging type of pediatric leukemia. Further, previously available methods to analyze chromatin structure such as ChIP-Seq

require large numbers of cells thus challenging a differential analysis of the various and rare subtypes of maturing T-cell precursors in the human thymus. Therefore, the development of ATAC-Seq which allows a highly reproducible and sensitive assessment of chromatin accessibility in small numbers of cells (Buenrostro *et al*, 2013) enabled us to answer fundamental questions such as how chromatin accessibility defines different maturation stages of T-cell development and at what point in this maturation hierarchy T-cell precursor ALLs originate. We report here that the physiological maturation of T cells is characterized by progressive condensation of the chromatin and a progressive decrease in the number of binding motifs for TFs. These findings are consistent with data obtained by quantitative high-resolution microscopy, which showed that embryonic stem cells contain significantly more euchromatin than hematopoietic stem cells, with a further reduction in mature cells (Ugarte *et al*, 2015).

Specifically, our data exhibit that the chromatin landscape of maturing healthy T cells is remodeled gradually reflecting the continuous transition from early double-negative to mature single-positive CD4$^+$ T helper cells and CD8$^+$ T cytotoxic cells. The higher fraction of OCRs with a closing profile than those with an opening profile (29% vs. 0.3%) in the course of maturation implicates that chromatin remodeling during T-cell maturation is mediated by condensing accessible regions instead of decondensing packed regions. FEA of the signature open chromatin regions revealed expected GO-terms which were almost exclusively related to T-cell biology (Dataset EV12). Notably, however, the signature peak with the highest PC1 contribution was identified in the body of *SAE1* which is a key enzyme in the process of SUMOylation. This finding implicates activation of SUMOylation in T-cell development and is consistent with a previous report linking activity of the SUMO-specific protease 1 (SENP1) to be essential for the development of early T- and B cells (Nguyen *et al*, 2008). Moreover, SAE1 is required for cell-autonomous definitive hematopoiesis, suggesting SAE1 importance for maintenance of hematopoietic stem/progenitor (Li *et al*, 2012). In this context, it is interesting to note that the analysis of chromatin accessibility reported here suggests a potential role of SAE1 in normal T-cell development.

In parallel to the condensation of the chromatin in the course of T-cell maturation, the number of open TF-binding motifs diminished gradually across the developmental stages. This global condensation of the chromatin is consistent with the previously observed shutdown of housekeeping genes during transition to the DP stages during murine thymocyte differentiation (Mingueneau *et al*, 2013). Specifically, the data presented here identified the TF SPI1 (PU.1), a pioneering TF during T-cell development (Ungerback *et al*, 2018) together with the small Maf proteins, Bach1/2, NFE2 and NF-E2 Related Factor 2 to be the most discriminating regulators of T-cell differentiation, displaying progressive losses during maturation. In the recently published results of the ImmGen Project, the expression of murine TFs was correlated with changes in chromatin accessibility of TF-binding sites. Thereby, a cis-regulatory atlas of 86 primary cell types spanning the mouse immune system was established (Yoshida *et al*, 2019). Comparison of TF-binding motifs, which distinguish differentiation stages, in human and mouse datasets showed that 26/50 of our top candidates are also differentially accessible in mouse immune cells, indicating a large degree of conservation of these regulatory factors between human and mice (Dataset EV5). The genome-wide chromatin accessibility atlas of human T cells that we have generated in this project will thus be a valuable resource for future studies of T-cell maturation and the dysfunction of T cells in human cancer.

The projection of OCRs from T-ALLs into the chromatin accessibility maps of human thymic T-cell precursors, combined with the deconvolution analysis, revealed that T-ALLs resemble immature T-cell precursors. All leukemias, even those that are categorized as mature by immunophenotyping, exhibit an immature pattern of chromatin structure and an early profile of accessibility of TF-binding sites. Moreover, the chromatin of the T-ALLs was significantly and substantially more accessible than that of the most immature of the sorted normal T-cell precursor population.

Based on the profiles of accessible TF-binding motifs, we identified two distinct groups of T-ALLs. The first group consisted of 3/19 patients who were characterized by highly accessible *SPI1*-binding sites, overexpression of *SPI1* when compared to bulk thymus paralleled by a particularly high contribution (> 60%) of cells deconvoluted to the early developmental stage DN2. For normal T-cell development to occur, *SPI1* downregulation is required at the double-negative stage when cells are committed to the T-cell lineage (Anderson *et al*, 2002b). Further, constitutive expression of *SPI1* in fetal thymic organ culture has been reported to result in reduced thymocyte expansion and blocked differentiation at the DN3 stage (Anderson *et al*, 2002b). Interestingly, fusions of *SPI1* have previously been detected in a small subgroup of pediatric T-ALL patients. The cells of these patients showed a double-negative (DN; CD4$^-$CD8$^-$) or CD8$^+$ single-positive (SP) phenotype and marked a subset of patients with a very poor outcome (Seki *et al*, 2017). We did not identify a fusion event explaining the high *SPI1* expression in the patients reported here, which indicates that an increased accessibility of *SPI1*-binding motifs resulting in its overexpression may be an alternative mechanism leading to the early developmental arrest. It will now be interesting to analyze the fusion-independent effect on prognosis and treatment resistance of an open *SPI1* chromatin structure and RNA expression in a clinically well documented larger cohort of patients.

The second larger group of 15/19 patients did not display hyperaccessible *SPI1*-binding sites but was characterized by a hyperaccessibility and expression of *DAB1*, which codes for an adaptor protein in the Reelin signaling pathway, inhibiting Notch-ICD degradation via Fbxw7 (Hashimoto-Torii *et al*, 2008) and activating PI3K and Akt (Jossin & Goffinet, 2007) in response to Reelin. Considering the overexpression of *DAB1* in the analyses reported here, it is remarkable that *DAB1* was not reported to be overexpressed in previous studies, possibly due to the lack of a contrast group of healthy T cells to detect differential expression. We verified our RNA-Seq-based findings with respect to *DAB1* using qPCR analyses of T-ALL cell lines, thymic cells and patient-derived T-ALL cells and recapitulated overexpression in Affymetrix microarray data. We have also identified a previously unrecognized *DAB1* transcript in T-ALLs. Future work will be required to investigate whether this novel transcript affects the viability or proliferation of leukemia cells. As for the role of *SPI1*, it will now be interesting to evaluate in a larger number of patients in how far *DAB1* hyperaccessiblity/overexpression impacts on prognosis and treatment resistance.

In conclusion, integrated analyses of chromatin accessibility and RNA expression implicate activation of the SUMOylation pathway in the maturation of thymic T-cell precursors and identify particularly

immature developmental stages to be the closest to pediatric T-ALLs, which are characterized either by activated *DAB1* or *SPI1* in our patient cohort.

# Materials and Methods

### Patients' clinical characteristics

The primary cells were obtained from patients recruited in ALL-BFM 2000, ALL-BFM-2009, CoALL03 and CoALL09 trials. For patients' clinical characteristics, see Dataset EV6.

The clinical trials including scientific analyses of the samples obtained from recruited patients have been approved by the relevant institutional review boards or ethics committees. Written informed consent had been obtained from all the patients and the experiments conformed to the principles set out in the WMA Declaration of Helsinki and the Department of Health and Human Services Belmont Report.

### Establishment of the patient-derived xenograft models

Patient-derived xenografts (PDX) were generated as described (Schmitz *et al*, 2011) by intrafemoral injection of $1 \times 10^5$ to $5 \times 10^6$ viable primary ALL cells in NSG (NOD.Cg-Prkdc^scid^Il2rgtm1Wjl/SzJ) mice. Transplanted mice were both male and female, aged 5–8 weeks. Animals were housed in individually ventilated cages with access to food and water *ad libitum*. Leukemia progression was monitored in the peripheral blood by flow cytometry using anti-mCD45, anti-hCD45, anti-hCD19, or anti-hCD7 antibodies. Cells had been harvested after engraftment reached 75% in the peripheral blood or mice health score reached either three at single item or the total score had reached five. T-ALL cells were collected from spleen and cryopreserved as described (Schmitz *et al*, 2011). Blast enrichment in the sample had been evaluated by flow cytometry using same antibody panel. Xenograft identity was verified by DNA fingerprinting using the commercial AmpFlSTR® NGM SElect kit. In vivo experiments were approved by the veterinary office of the Canton of Zurich, in compliance with ethical regulations for animal research.

### Thymic cell collection

Thymic samples were collected from otherwise healthy children who underwent cardiac surgery. Cell Sorting and ATAC-Seq experiments using thymic samples was approved by the ethics committees of Heidelberg University Hospital and EMBL Heidelberg. Written informed consent was obtained from the donors' parents.

For disaggregation, collected thymus was cut into small pieces and placed between two nylon meshes in a petri dish. Complete RPMI 1640 media (GIBCO, life technologies) were added to the dish, and thymocytes were released by gently pushing the tissue with a syringe plunger. The cell suspension was filtered with a 40 μm cell strainer (Falcon, Corning) and labeled as described below.

### Multicolor fluorescence activated cell sorting

Isolated cells were incubated with human Fc Block for 15 min at 4°C, washed, and stained 30 min at 4°C with an antibody mix containing APC-conjugated CD7 (M-T701; 1:100), BV421-conjugated CD34 (581; 1:50), PE-Cy7-conjugated CD38 (HIT2; 1:10,000), PE-conjugated CD1a (HI149; 1:200), APC-Cy7-conjugated CD3 (SK7; Leu-4; 1:100), FITC-conjugated CD4 (RPA-T4; 1:100), and BV605-conjugated CD8 (SK1; 1:200). All antibodies and Fc Block were purchased from BD Biosciences.

Stained samples were analyzed, and several populations of early T-cell precursors and mature T cells were identified. Seven of these populations were sorted following the gating strategy depicted in Appendix Fig S12, using a BD FACSAria™ Fusion cell sorter (BD Biosciences) equipped with an 85 μm nozzle. Dead cells were excluded by adding 7-AAD (7-aminoactinomycin D; BD Biosciences) to the cell suspension, and doublets were carefully removed by plotting SSC-area vs. SSC-width and FSC-height vs. FCS-area; cells with increased width/area were not considered.

A fraction of the sorted cells was re-analyzed to verify the purity of each population. Purity was > 95% for the seven sorted fractions. Post-acquisition analysis was done with FlowJo software 10.0.8 (Tree Star, Inc.). The details of the fractions of each population identified in six donor thymi are summarized in Appendix Table S1.

### ATAC sequencing

Analysis as described in the PDX models recapitulate the genetic and epigenetic landscape of pediatric T-cell leukemia (Richter-Pechanska *et al*, 2018). OCRs were assigned to the TSS category if they fell into the ± 1 kb window of the transcription start site of a CCDS (The Consensus Coding Sequence, (Pruitt *et al*, 2009)) gene. All other OCRs, outside the ± 1 kb window, were assigned to the non-TSS/distal category.

Peak categorization into four patterns (increasing, decreasing, fluctuating, and steady) was based on the standard deviation in the normalized read counts and the accessibility profile of a peak in the course of maturation: increasing: $SD \geq 12$ and RCs of consecutive stages increase, decreasing: $SD \geq 12$ and RCs of consecutive stages decrease, fluctuating: $SD \geq 12$ and RCs going up/down during development, steady: $SD \leq 12$ or $SD \geq 12$ and difference between RCs of consecutive stages between $-24$ and 24.

### Generation of signature matrix

One-versus-all testing was used to collect significantly different peaks for each of the developmental stages. The thymus dataset of 58,294 distal peaks was used as input for differential accessibility analysis by DESeq2 (LFC = 0, FDR = 0.3) (Love *et al*, 2014). For every developmental stage, the $n$-peaks were selected with the largest LFC and added to a signature peak list keeping track of unique peaks. The base number $n$ was optimized for the lowest Cohen's Kappa value of the signature matrix; with a parameter sweep from 300 to 800 in intervals of 10. Variance stabilizing transformation (VST) was used to normalize the read counts in the signature matrix. For a base number $n = 790$, the final signature matrix after removing duplicates contained 2,823 peaks.

Cross-validation was performed using a standard leave-one-out approach. Signature matrices with the same base number $n$ were created from all samples excluding one, and this is repeated as many time times as there are samples. Instead of recollecting peaks, a read count matrix was used with the column of that sample

removed to ensure the same joined set of peaks was used throughout the cross-validation process. Subsequently, the excluded sample was deconvoluted with CIBERSORT (Newman et al, 2015) using the signature matrix constructed without that sample. For every sample, the test was marked as a pass/fail based on a match between the most dominant population according to the deconvolution and the developmental stage the sample was sorted as.

The signature matrix of the seven sorted developmental stages (2,021 peaks, $n = 375$) performed poorly in leave-one-out cross-validation. Only 25 out of 37 (68%) samples could be identified as the stages as they were sorted as. Only the CD4s and CD8s were identified correctly. However, the double-positive populations (DPCD3$^-$ and DPCD3$^+$) and the double-negative populations (DN2, DN3 and ISP) could not be distinguished.

In a time course experiment, we showed that high similarity between two DP populations is caused by the internalization of CD3 surface markers of DPCD3$^+$ cells during long periods of sorting (Appendix Fig S4). As a result of internalization, the cells which are actually CD3$^+$, fell into the CD3$^-$ cells' gate, which explained why DPCD3$^-$ populations had DPCD3$^+$ contribution in deconvolution analysis. Merging of two DP populations to generate signature matrix improved the cross-validation results to 84%, yet the uncertainties in the classification of double-negative populations remained.

Several efforts were made to improve the classification of DN populations. Merging the DN2 and DN3 populations resulted in a correct rate of 92%, excluding DN3 populations from the analysis resulted in 97%, and merging DN3 and ISP populations resulted in 95% correct rate (30/32; Appendix Fig S3A and B). Instead of excluding the DN3 population, this population was therefore merged with the ISP population.

Principal component analysis was performed on the signature peaks to sort the peaks by importance or contribution using the rotations of PC1 and PC2.

## Functional enrichment analysis

Functional enrichment analysis of signature OCRs of five developmental groups was performed by Genomic Regions Enrichment of Annotations Tool (GREAT) version 4.0.4 (McLean et al, 2010). OCRs were assigned to the group with the highest VST normalized read count. Each developmental group contributed different numbers of peaks to the final signature matrix (Appendix Fig S5). FEA did not yield any results when it was run separately for each of the five groups because of the low number of peaks per group. Therefore, peaks contributed by DN2 and DN3.ISP stages and SPCD4$^+$ and SPCD8$^+$ stages were combined to obtain three groups of DN, DP, and SP populations when running GREAT analysis.

## Motif enrichment analysis

The thymus dataset of 58,294 distal peaks was annotated using Alfred (Rausch et al, 2019) with motifs from the non-redundant JASPAR CORE vertebrate PFM catalog (Khan et al, 2018). To define expressed genes, a threshold of > 0.5 FPKM was used as defined by the ENCODE project. Only the motifs of TFs that were identified to be expressed in T cells were considered. Motif

**Table 1. Example contingency table for Fisher's exact test to calculate BACH1 enrichment in DN2.**

| | DN2 | Other groups (excluding DN2) |
|---|---|---|
| BACH1 | Occurrences of BACH1 in peaks of DN2 cells | Sum of BACH1 occurrences in groups other than DN2 |
| Other motifs (excluding BACH1) | Sum of motif occurrences other than BACH1 in DN2 cells | Sum of motif occurrences other than BACH1 in groups other than DN2 |

positions were intersected with footprint positions inferred by HINT-ATAC (Li et al, 2019) using the distal peaks (non-TSS peaks) of every sample. These footprints represent possible TF-binding sites which are dips within the accessible peaks. For downstream analyses, the motif occurrences within a footprint score > 25 were considered. This cutoff is approximately the average score of footprints in most samples, and these footprints were also verified with manual inspection. Since many TFs are highly similar, we pre-filtered motifs that shared greater than 90% of their mapping locations.

Fisher's exact test was used to calculate whether a motif is enriched within a developmental group. First, a count matrix was constructed with rows for every motif and columns for each group. Hereto, the motif occurrences were summed per motif per sample and subsequently per group. These counts were used to fill out the contingency table for each motif-group pair. An example of a contingency table is given in Table 1 for the enrichment of BACH1 in the DN2 group. The P-value was adjusted for multiple testing with Bonferroni and FDR (P.adjust R function).

For visualization in Fig 3A, TFs which have < 10 standard deviation in the motif counts were excluded first. Then, the top 50 significant TFs ($P_{adj} < 0.05$) which exhibit the highest standard deviation in the odds ratios were selected to highlight motifs that discriminate between developmental stages.

## Deconvolution analysis

The deconvolution algorithm CIBERSORT (Newman et al, 2015) was used to deconvolute the T-ALL samples. An in-house generated signature matrix of T-cell developmental stages was used as input for the algorithm. The reads in the 2,823 signature peaks in T-ALL samples were counted and normalized with variance stabilizing transformation. CIBERSORT predicted the contribution of each developmental stage as a fraction with a correlation and root mean square error to display the goodness of fit.

## Differential analyses

Global differential accessibility analysis of all T-ALLs and sorted populations was run by DESeq2 (LFC = 0.5, FDR = 0.05) (Love et al, 2014) using all TSS and distal peaks. Mean read counts were used for 10/19 T-ALL samples having biological replicates. Global differential expression analysis of all T-ALLs and three bulk thymi was run by DESeq2 (LFC = 0.5, FDR = 0.05) (Love et al, 2014). Genes with low expression were excluded using a threshold of < 20 mean read count. Mean read counts were used for 7/19 T-ALL samples having biological replicates.

## Integration of ATAC- and RNA-Seq data

To account for patient heterogeneity (Fig 5B) and to quantify recurrence of dysregulation of a gene, differential analysis of ATAC- and RNA-Seq data described above was done on a per patient basis with a parameter change: genes with < 5 mean RNA read count were excluded. ATAC peaks were annotated using Alfred's (Rausch *et al*, 2019) annotate subcommand using $-d$ 10,000 bp, resulting in zero or more genes assigned to a peak. The 1000 Genomes reference genome (*Homo sapiens GRCh37$^+$ decoy sequences)* was used for alignment and peak calling. Differential peaks and genes were intersected based on this annotation, resulting in the association of one peak with multiple genes and vice versa. The combination of all patients' results was divided into 4 quartiles as shown in Fig 5c: Q1 (logarithmic fold change [lfc] RNA > 0, lfc_ATAC > 0), Q2 (lfc_RNA > 0, lfc_ATAC < 0), Q3 (lfc_RNA < 0, lfc_ATAC < 0), and Q4 (lfc_RNA < 0, lfc_ATAC > 0). To quantify the recurrence of a dysregulation of a gene, each gene–patient pair was counted once per quartile.

## Total RNA-Seq

Total RNA was extracted using TRIzol (Invitrogen Life Technologies). RNA was than treated with TURBO DNase (Thermo Fisher Scientific, Darmstadt, Germany) and purified using RNA Clean & Concentrator-5 (Zymo Research, Freiburg, Germany). A minimal RIN (RNA Integrity Number) of 7 as measured using a Bioanalyzer (Agilent, Santa Clara, CA) with the Agilent RNA 6000 Nano Kit was required for the sample to be sequenced. Cytoplasmic ribosomal RNA was depleted by Ribo-Zero rRNA Removal Kit (Illumina, San Diego, CA), and the libraries were prepared from 1 μg of RNA using the TruSeq RNA Library Prep (Illumina, San Diego, CA) at the Genomics Core Facility of the EMBL, Heidelberg. Six RNA samples were pooled and sequenced on one Illumina NextSeq 500 (Illumina, San Diego, CA, USA) lane in 75 bp paired-end mode. Adapters were removed using cutadapt, and upon classification, only the human reads were aligned using STAR aligner (Dobin *et al*, 2013). Gene fusion discovery in RNA-Seq data was performed using Arriba and deFuse algorithms. Genes with less than 20 reads in all samples were ignored. FPKM (Fragments Per Kilobase of transcript per Million mapped reads) normalized RNA read counts are in Dataset EV13.

## qPCR and Sanger sequencing

RNA of the five T-ALL cell lines and four T-ALL patients were isolated by either Qiagen AllPrep DNA/RNA/Protein Mini Kit or by the standard phenol-chloroform extraction protocol. RNA concentration was assessed by NanoDrop and 1 μg of RNA is used for cDNA synthesis. Forward and reverse primer sequences can be found in Table 2.

**Table 2. Primer sequences used for qPCR and Sanger sequencing.**

|  | Forward | Reverse |
| --- | --- | --- |
| qPCR | ACCAGCGCCAAGAAAGACTC | TGTTCTCCTTTGGAACGAGCG |
| Sanger sequencing | TACACAGCTTGTTCACACTGC | GGCCCTTGGGAGCTTTTAGA |

### The paper explained

#### Problem

Precursor T-cell leukemias in children are a particular challenge, because relapses are exquisitely treatment resistant and because the mechanisms of leukemogenesis and progression are poorly understood. Recent genomic analyses revealed that mutations in epigenetic modulators are common in this type of leukemia suggesting that epigenetic mechanisms may play an important part in the process of leukemogenesis. We have thus aimed at defining epigenetic differences between normal and leukemia cells by performing analyses of chromatin structure and the accessibility of TF-binding motifs in normal human T-cell precursors and in pediatric precursor T-cell leukemias (T-ALL).

#### Results

We found that chromatin accessibility decreased gradually in the course of maturation and that pediatric T-ALLs highly resembled the most immature developmental stages of healthy T cells in terms of both, genome-wide chromatin accessibility in general and accessibility of TF-binding motifs in particular. The T-ALL chromatin landscape was substantially more accessible than that of even the most immature of the healthy T-cell precursor populations.

Integration of epigenomic (ATAC-Seq) and transcriptomic (RNA-Seq) analyses identified differentially accessible chromatin regions surrounding differentially expressed genes in T-ALLs in comparison to normal thymic precursors. *DAB1*, a gene that has previously been noted to be highly expressed during neural development but not in leukemia, was the most recurrently dysregulated gene in the T-ALLs and also exhibited the highest capacity to distinguish between leukemia cells and healthy T-cell precursors. In a smaller subgroup of patients showing the most immature epigenetic signature, we identified accessibility and overexpression of *SPI1(PU.1)* as a defining feature thus implicating an important role of this well-known hematopoietic transcription factor in this specific subtype of T-ALL.

#### Impact

Our results indicate a progressive condensation of the chromatin in the course of human T-cell maturation and, for the first time, document the changes in the chromatin accessibility profile of T-ALLs in comparison to normally differentiating T-cell precursors.

We conclude that the chromatin of T-ALL is substantially more open than that of healthy thymic precursors and that leukemia-specific alterations of chromatin conformation reveal previously unrecognized mechanisms of pediatric T-ALL leukemogenesis.

### Software and bioinformatical tools

Graphical representation and statistics were done using: R (R Core Team, 2017) and GraphPad Prism version 6.00 for Windows (La Jolla, California, USA, www.graphpad.com).

Functional enrichment analyses for hyper-/hypo-accessible ATAC-regions and their graphical representation were done using GREAT version 4.0.4 (Genomic Regions Enrichment of Annotations Tool) (McLean *et al*, 2010).

Visualization of publicly available data of Affymetrix microarrays were done using R2: Genomics Analysis and Visualization Platform (http://r2.amc.nl).

Differential expression analysis of publicly available microarray dataset of sorted human thymocyte populations was performed with GEO2R (Barrett *et al*, 2013).

## Data availability

Sequence data have been deposited at the European Genome-phenome Archive (EGA, http://www.ebi.ac.uk/ega/), which is hosted by the EBI, under accession number EGAS00001003248 (https://ega-archive.org/studies/EGAS00001003248).

**Expanded View** for this article is available online.

## Acknowledgements

This work was supported by Deutsche José Carreras Leukämie Stiftung, by the Manfred Lautenschläger Stiftung and by the Aktion für krebskranke Kinder e.V. Heidelberg, and by the Baden-Württemberg-Stiftung. We thank EMBL Genomics Core Facility for the preparation and sequencing of RNA-Seq libraries and for the sequencing of ATAC and WES libraries. We thank all the patients and thymus donors who participated in the study and their families. Open access funding enabled and organized by Projekt DEAL.

## Author contributions

BE-U designed and performed the experiments, performed the bioinformatic analyses and wrote the manuscript, JBK designed the research, contributed to the interpretation of the results and wrote the manuscript, TR performed bioinformatic analyses, aided in interpreting the results and wrote the manuscript, PR-P aided in the bioinformatics analyses and interpreting the results and wrote the manuscript, IAEMVB performed the bioinformatic analyses of ATAC-Seq, VF and BB established the PDX models, DOR and MP aided in developing FACS staining panel and performing sorting, VB run ATAC-Seq libraries, MS, MSc, GC, GE, KB, RK-S & CE provided patients samples, TL and MG provided thymic samples from healthy donors, SMW contributed to the design of the research and interpretation of the results, J-PB contributed to the design of the research and supervised the establishment of the PDX models, MUM contributed to the analysis of the results and to the writing of the manuscript, JOK and AEK designed the research, supervised the project, interpreted the data and wrote the manuscript. All authors reviewed and contributed to the final manuscript.

## Conflict of interest

The authors declare that they have no conflict of interest.

## For more information

https://github.com/tobiasrausch/ATACseq

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
