## [Review Process File · EMBO Molecular Medicine]

Chromatin accessibility landscape of pediatric T-lymphoblastic leukemia and human T-cell precursors

Büşra Erarslan-Uysal, Joachim Kunz, Tobias Rausch, Paulina Richter-Pechanska, Ianthe Belzen, Viktoras Frismantas, Beat Bornhauser, Diana Ordonez, Malte Paulsen, Vladimir Benes, Martin Stanulla, Martin Schrappe, Gunnar Cario, Gabriele Escherich, Kseniya Bakharevich, Renate Kirschner-Schwabe, Cornelia Eckert, Tsvetomir Loukanov, Matthias Gorenflo, Sebastian Waszak, Jean-Pierre Bourquin, Martina Muckenthaler, Jan Korbel, and Andreas Kulozik

DOI: 10.15252/emmm.202012104

Corresponding author(s): Andreas Kulozik (andreas.kulozik@med.uni-heidelberg.de), Jan Korbel (korbel@embl.de)

Review Timeline:	Submission Date:	1st Feb 20
	Editorial Decision:	18th Feb 20
	Revision Received:	18th May 20
	Editorial Decision:	3rd Jun 20
	Correspondence:	9th June 20
	Revision Received:	19th Jun 20
	Accepted:	24th Jun 20

Editor: Jingyi Hou

Transaction Report:

Thank you for the submission of your manuscript to EMBO Molecular Medicine. We have now heard back from the two referees whom we asked to evaluate your manuscript. As you will see from the reports below, the referees acknowledge the potential interest of the study. However, they also raise a series of concerns on your work, which should be convincingly addressed in a major revision of the present manuscript. In particular, a direct comparison of the presented human T-cell data with existing mouse datasets needs to be performed. Further, additional experiments and analysis are required to strengthen the proposed functional roles of DAB1 and SUMO1 in T-ALL.

All other issues raised by the referees need to be satisfactorily addressed as well. Please note that EMBO Molecular Medicine strongly supports a single round of revision and that, as acceptance or rejection of the manuscript will depend on another round of review, your responses should be as complete as possible.

Please also contact us as soon as possible if similar work is published elsewhere. If other work is published we may not be able to extend the revision period beyond three months. Please read below for important editorial formatting and consult our author's guidelines for proper formatting of your revised article for EMBO Molecular Medicine.

REFEREE REPORTS

***** Reviewer's comments *****

Referee #1 (Comments on Novelty/Model System for Author):

Primary human cells and leukemia xenografts are ideal for understanding human disease, but comparison of human datasets to existing high-quality mouse RNA-Seq and ATAC-Seq datasets could have enhanced the impact of this study.

Referee #1 (Remarks for Author):

General comments:

The authors generate chromatin accessibility (ATAC-Seq) and RNA-Seq data from sorted subpopulations of human thymic T-cell progenitors. Subsequent analysis of this data is aimed at analysis of differential accessibility peaks (nucleosome-free regions, NFRs) in order to identify transcription factors and genes with differential roles across stages of T-cell development. The authors also use a classifier trained on chromatin accessibility data to sub-classify previously published T-ALL xenograft ATAC-Seq datasets according to their corresponding normal stages of development. Informed by their normal T-cell analysis, the authors nominate differentially regulated genes of potential significance in their T-ALL datasets.

The strengths of this work include the profiling of well-defined human cell populations, which are much more difficult to obtain and work with than cell populations from model organisms, and which can provide unique insights regarding human disease. The authors' approach to validating the ATAC-Seq data and classifier peak set, which identified some sorted subpopulations that could not be accurately distinguished by the data obtained (due to technical limitations) reflects an appropriate level of rigor.

Weaknesses of this observational work include an effort to nominate specific genes as having novel developmental or oncogenic function in the absence of sufficient supporting data on the basis of being near the most significant differential chromatin peak.

One important challenge in TF motif enrichment analysis is that motifs for many TFs (usually within the same family) are extremely similar and cannot be distinguished by DNA sequence. For example, the author's 'top 50' TF motifs include many ETS factors with essentially indistinguishable motifs (although the PU.1 motif is divergent). The authors should group highly similar TF motifs (defined by hierarchical trees available in JASPAR) to show which of their significant TF motifs are independent of others. This is particularly important since the authors' analysis highlights putatively significant TFs with previously unknown function in T-cell development (based on expression and motif enrichment), but motif enrichment for these proposed novel regulatory factors might be entirely explained by known factors with similar / identical binding motifs. "Novel" factors with ambiguous / redundant motifs with known factors should therefore be separately designated.

It is surprising that the authors do not attempt to directly compare findings from their analysis of human T-cell ATAC-Seq and RNA-Seq data with corresponding high-quality public datasets

obtained in mouse (e.g. by the Immgen project). Such an analysis could significantly help support some of the authors' claims, since most major regulatory factors would be expected to be conserved in humans and mice, while allowing them to emphasize any findings that are unique to humans, and may therefore reflect the unique value of this data in understanding human T-cell developmental and leukemia. The authors might at least mention the existence of these datasets and suggest that future mouse-human comparisons would be a valuable use of this resource.

Specific comments:

Introduction: It may be more accurate to say that T-cell leukemias were 'historically' subclassified in analogy to T-cell maturation, as the surface marker / maturation stage- based classification mentioned by the authors (Bene et al 1995) is no longer used in clinical diagnostic practice or widely in research. It has generally been superseded by the oncogene-based classification (TAL / LMO / TLX / HOXA etc) plus the ETP-T-ALL maturation category, which overlaps with the earliest stages of the previous classification.

Suppl figure 1: It is surprising that such a small fraction of ATAC-Seq peaks are in intergenic and intronic regions in all populations. Most distal enhancers are located in these compartments, and they comprise a high fraction of peaks in most high-quality ATAC-Seq datasets from most cell types. It is unclear to me what the authors mean by "non-coding" as a category of genomic region exclusive of intergenic / TSS / TTS / intron / 5' UTR / 3' UTR, all of which are themselves non-coding regions. Given that TSS sites were excluded for most of the analysis, it seems that these "non-coding" peaks, as well as TTS sites make up the majority of the peaks analyzed for the differential analysis.

The mention of using CIBERSORT to deconvolute T-ALL on lines 161-162 (in the context of figure 2) is confusing, as the actual analysis of T-ALL samples does not occur until a later section (figure 4). It should be clarified that CIBERSORT is being used to validate the signature matrix for normal cells only in this part of the paper.

Figure 2B: What is the significance of the triangles marked "known regulatory element binding site" (explained in the legend as "known promoter or enhancer regions")? Through what dataset were these "known enhancers" defined?

Figure 3a appears to be a schematic diagram explaining the rationale of the ATAC-Seq footprinting analysis rather than actual data, but the sizes depicted are surprising. Distal enhancer NFR peaks detected by ATAC-Seq are typically on the order of 200-300bp, while high-quality TF footprints should be closer to the size of DNA contacted by the TF itself (8-20 bp), though some protected regions are larger due to TF complexes. If the footprints detected in these datasets are really 70 bp in average size, this analysis may just be more accurately defining NFRs, rather than true individual TF footprints. Footprinting analysis with ATAC-Seq has been controversial, as it requires very deeply sequenced and high-quality data, as well as analytic methods that can account for the insertion biases of TN5 transposase (such as the HINT-ATAC method used by the authors). In order for the reader to understand the value of this analysis on this dataset, the authors should show example loci with actual MACS2 / IDR ATAC-Seq peak calls, HINT-ATAC footprinting calls, and example TF motif instances. They should also show global TF motif enrichment analysis performed on the peak calls alone versus motifs occurring only within footprints, in order to confirm that the footprinting analysis adds value.

More detail is required to describe the motif annotation performed with ALFRED and the JASPAR

database. Was the analysis restricted to JASPAR CORE vertebrate PFM's? Were peaks / footprints just overlapped with pre-computed genome-wide motif calls from JASPAR, or were custom thresholds or motif identification criteria used?

For figure 3B, the star classification system for literature review requires further explanation. What is the difference between "T-cell development related" and "stage-specific activity" and how are these concepts exclusive of one another?

The authors should discuss the degree to which global differences between T-ALL xenograft and purified normal thymic T cell precursors, such as the greater number of peaks seen in T-ALL samples, might be due to technical issues such as batch effects and / or processing time.

In figure 5C, it is unclear to me what normal progenitor dataset is being used to identify differential gene expression and chromatin accessibility in normal vs. T-ALL - is the comparison to bulk thymus RNA / ATAC data, or to the sorted sub-populations (and if so, which ones)? Both comparisons are mentioned in the methods.

The author's emphasis on DAB1 as a putative novel essential gene in "leukemia" based on having a strong accessibility peak in T-ALL versus normal T-cell progenitors is not well supported by the presented evidence. Despite the finding that DAB1 is upregulated in some T-ALL, the authors note that the absolute quantity of DAB1 transcripts is low (reflected in low FPKM), and it is not clear why the large size of the transcript (which produces a larger absolute number of reads and makes it more readily detectable by RNA-Seq despite a low absolute number of transcripts) is indicative of greater biological significance than its low expression would suggest. The authors reference public DepMap RNAi screen data performed on 25 leukemia cell lines, only 2 of which are T-ALL. The average DEMETER dependency score of -0.21 across "leukemia" is not an especially strong effect. Unmentioned by the authors is the fact the two T-ALL cell lines show among the weakest effects of any leukemia cell line. (both > -0.1, likely insignificant). Furthermore, more specific CRISPR screen data is also available through DepMap, and shows DAB1 dependency scores near zero for three T-ALL cell lines, and a near-zero average score for leukemia cell lines in general, indistinguishable from other cancer types (with the possible exception of peripheral nerve tumors). In the absence of other experimental support for a functional role of DAB1 in T-ALL, this emphasis should be substantially reduced.

Referee #2 (Remarks for Author):

This is an interesting study of T-ALL and normal T-cell development based on ATAC-sequencing, transcription factor motif analyses and gene expression data. Overall, the authors reveal the open/closed chromatin regions in T-cell subsets and T-ALL samples by ATAC-seq and use that information to compare T-ALL with T-cell subsets. They discover that T-ALL cells are most similar to immature T-cells in DN2/DN3 stage. Based on transcription factor motifs present in the accessible regions, they estimate which transcription factors are most likely implicated in the regulation of gene expression. Moreover, in additional analyses with gene expression data, the authors identify DAB1 as a gene that is upregulated in T-ALL.

This work is of interest and provides novel views on the regulation of T-cell and T-ALL development, but remains largely descriptive as no genes are inactivated or overexpressed to study the observations made in more detail (except for the use of publicly available loss-of-function screen data for DAB1).

Major comments:

- The abstract is misleading and seems to indicate that the authors study the cell of origin of T-ALL, which is not done at all. Moreover, the emphasis on SUMO1 and DAB1 is overestimating the potential role that these factors play in T-ALL as there is no proof for their importance (only expression is documented).
- Figure 1: rather than to only show the total number of ATAC-seq peaks in figure 1, and to show that the total number is decreasing upon differentiation of T-cells, it would be also informative to show the number of regions that become less accessible and also the number of regions that become more accessible - if there are such regions.
- For the transcription factors shown in figure 3: are these transcription factors expressed at these stages of T-cell development? Such expression data should be taken into account if possible.
- If the authors hypothesize that SUMO1 is important: is there also publicly available loss-of-function screen data for this gene (as used for DAB1)? This is currently all hypothetical.
- For DAB1: the authors use information from a publicly available loss-of-function screen. Can a more broad analysis be performed where the ATAC-seq/gene expression data is compared with the loss-of-function screen for all the top candidate genes that are similar to DAB1 (specifically expressed in T-ALL and in accessible regions) ?

Minor comments:

- The authors state "Chromatin accessibility as measured by the number of OCRs decreased progressively with maturation, particularly at promoter and non-coding regions (Suppl. Fig. 1)." Can the authors document this more detailed towards promoter and known transcriptional regulatory regions (for example H3K27ac regions)? Also, what exactly are the regions defined in Suppl fig 1 (for example 'transcription termination site' - how are these regions defined and how large are these regions ? 1 kb around termination site or what exactly ?); so it remains poorly specified where the OCRs decrease exactly.
- The authors find that the expression of DAB1 is 41-times higher in the T-ALLs than in bulk thymus. It would be more appropriate to check DAB1 expression in the different normal subsets of the thymus, since each of the subsets may show different DAB1 levels - bulk thymus expression level is not very informative, especially since the authors show that T-ALL is more comparable to DN2/DN3 stages than to the more differentiated stages or to the DP stage which is the most frequent cell type in thymus.

Erarslan-Uysal *et al.*, point-by-point response to reviewers

Referee #1 (Comments on Novelty/Model System for Author):

Primary human cells and leukemia xenografts are ideal for understanding human disease, but comparison of human datasets to existing high-quality mouse RNA-Seq and ATAC-Seq datasets could have enhanced the impact of this study.

Response: We thank this reviewer for this helpful suggestion. We have now compared existing mouse ATAC-Seq data and specifically the accessibility of transcription factor (TF) binding motifs in open chromatin regions (OCR) as published in the ImmGen Project (Yoshida *et al.*, 2019), showing that half of the differentially accessible TF-binding sites that we report in Fig 3B overlap between mouse and human thymocytes at different stages of development. We have now updated **Dataset EV5** to indicate TFs which exhibit differential accessibility both in human and mouse thymic subpopulations and included this new comparison in the Discussion section of the manuscript (**line 447**). In our study, we did not isolate RNA from thymic subpopulations in sufficient quantities for sequence analysis and are, therefore, unable, to compare RNA-Seq data from human and mouse studies.

In the recently published results of the ImmGen Project the expression of murine transcription factors were correlated with changes in chromatin accessibility of TF binding sites. Thereby, a cis-regulatory atlas of 86 primary cell types spanning the mouse immune system was established (Yoshida *et al.*, 2019). Comparison of differential TF binding motifs in human and mouse datasets showed that 26/50 of our top candidates are also differentially accessible in mouse immune cells, indicating a large degree of conservation of these regulatory factors between human and mice (Dataset EV5). The genome-wide chromatin accessibility atlas of human T-cells that we have generated in this project will thus be a valuable resource for future studies of T-cell maturation and the dysfunction of T-cells in human cancer.

Considering the suggestion of this reviewer to secure that high-quality primary data serve as an input for the analysis, we have performed a comparison of quality measurements with publicly available datasets of AML and sorted healthy hematopoietic cells (Corces *et al.*, 2016). We found that there are no statistically significant differences in the number of filtered peaks and in the fraction of reads mapping to contaminating mitochondrial genomes (**see boxplots below**). Therefore, we conclude that the quality of our data is comparable to other studies who analyzed human hematopoietic cells.

Referee #1 (Remarks for Author):

General comments:

The authors generate chromatin accessibility (ATAC-Seq) and RNA-Seq data from sorted subpopulations of human thymic T-cell progenitors. Subsequent analysis of this data is aimed at analysis of differential accessibility peaks (nucleosome-free regions, NFRs) in order to identify transcription factors and genes with differential roles across stages of T-cell development. The authors also use a classifier trained on chromatin accessibility data to subclassify previously published T-ALL xenograft ATAC-Seq datasets according to their corresponding normal stages of development. Informed by their normal T-cell analysis, the authors nominate differentially regulated genes of potential significance in their T-ALL datasets.

Response: We do appreciate the summary of our experimental strategy, although we would like to emphasize out that the majority of the ATAC-Seq data, in total 29 T-ALL samples of 19 patients, have been generated for this publication (only 7 samples of 4 patients have been reported previously). We have now updated the text of the manuscript (line 270) and the legend of Fig 4A accordingly:

Seven out of 29 PDX models (corresponding to 4/19 patients with T-ALL; Fig 4A) were previously shown to recapitulate the chromatin accessibility landscapes of primary leukemia samples and, in comparison to primary patient samples, to yield technically comparable material resulting in higher TSS enrichment scores of the ATAC library (pval=0.03; Wilcoxon test; (Richter-Pechanska et al., 2018)).

Fig 4A: T-ALL cells of 19 patients were engrafted into mice, cryopreserved after harvesting and enriched for viable cells after defrosting. Then, samples were subjected to ATAC-sequencing. Patients P7, P8, P10, P59 were published previously (Richter-Pechanska et al., 2018).

Second, as outlined above, because of the very small numbers of isolated cells in some thymic subfractions we could not isolate sufficient amounts of RNA from the sorted human

thymic subpopulations and therefore performed RNA-Seq analyses on bulk thymus. To avoid misunderstanding, we have now updated the legend of **Fig 5C**:

Fig 5C: log₂ fold changes (LFC) of 292 genes which are differentially expressed (x-axis, RNA-Seq: non-sorted bulk thymi vs. T-ALLs) and differentially accessible (y-axis, ATAC-Seq: sorted T-cell populations vs. T-ALLs) in T-ALLs in comparison to healthy T-cell precursors.

The strengths of this work include the profiling of well-defined human cell populations, which are much more difficult to obtain and work with than cell populations from model organisms, and which can provide unique insights regarding human disease. The authors' approach to validating the ATAC-Seq data and classifier peak set, which identified some sorted subpopulations that could not be accurately distinguished by the data obtained (due to technical limitations) reflects an appropriate level of rigor.

Response: We thank the reviewer for acknowledging the rigor and value of our data.

Weaknesses of this observational work include an effort to nominate specific genes as having novel developmental or oncogenic function in the absence of sufficient supporting data on the basis of being near the most significant differential chromatin peak.

Response: In order to address the above comment we have now performed new experiments elucidating the specific role of the top candidate *DAB1* that emerged to be most recurrently overexpressed and hyper-accessible in the T-ALLs when compared to the normal thymic precursors. The combination of ATAC-Seq and RNA-Seq together with a *DAB1* exon usage analysis revealed that the leukemias overexpress a previously unknown isoform of *DAB1* including additional exons. This discovery indicates that the integration of ATAC-Seq and RNA-Seq data can improve the power to uncover leukemia-specific abnormalities that were not called previously by RNA-Seq analyses alone. We now show these data in the new **Fig EV5**, referred to in **line 363** and in the Discussion section (**line 492**):

Further, we discovered that T-ALLs express a novel transcript of *DAB1* that is not annotated in the GTEx database but expressed in T-ALLs (Fig EV5). Expression of exons 10-15, which are not normally co-expressed, was validated by RT-PCRs and Sanger sequencing in two patients (P7 and P9) and three T-ALL cell lines (Fig EV5). The integration of ATAC-Seq and RNA-Seq data combined with an exon usage analysis thus revealed that the leukemias overexpress a previously unannotated isoform of *DAB1* including additional exons. This discovery indicates that the combination of methodologies used here is sensitive to uncover leukemia-specific abnormalities that were not called by previous RNA-Seq analyses alone. Future work will be required to investigate whether this previously unrecognized transcript of *DAB1* affects the viability or proliferation of leukemia cells in T-ALL.

We have also identified a previously unrecognized *DAB1* transcript in T-ALLs. Future work will be required to investigate whether this novel transcript affects the viability or proliferation of leukemia cells.

One important challenge in TF motif enrichment analysis is that motifs for many TFs (usually within the same family) are extremely similar and cannot be distinguished by DNA sequence. For example, the author's 'top 50' TF motifs include many ETS factors with essentially indistinguishable motifs (although the PU.1 motif is divergent). The authors should group highly similar TF motifs (defined by hierarchical trees available in JASPAR) to show which of their significant TF motifs are independent of others. This is particularly important since the authors' analysis highlights putatively significant TFs with previously unknown function in T-cell development (based on expression and motif enrichment), but motif enrichment for these proposed novel regulatory factors might be entirely explained by known factors with similar / identical binding motifs. "Novel" factors with ambiguous / redundant motifs with known factors should therefore be separately designated.

Response: We thank the reviewer for raising this important point. We have annotated OCRs with TF motifs from the non-redundant JASPAR CORE vertebrate PFM catalog (Khan et al., 2018) using Alfred (Rausch, Fritz, Korbel, & Benes, 2018) and then filtered motifs that shared greater than 90% of their mapping locations. Additionally, only the motifs of TFs which were identified to be expressed in T-cells are considered. We therefore already grouped highly similar motifs by means of picking a representative TF motif out of every cluster where the corresponding TF is expressed in T-cells. We have now explained these steps in more detail in the Methods section of the manuscript (**line 601**):

The thymus dataset of 58,294 distal peaks was annotated using Alfred (Rausch, Fritz, Korbel, & Benes, 2018) with motifs from the non-redundant JASPAR CORE vertebrate PFM catalog (Khan et al., 2018). Only the motifs of TFs that were identified to be expressed in T-cells were considered. To define expressed genes a threshold of >0.5 FPKM was used as defined by the ENCODE project. Motif positions were intersected with footprint positions inferred by HINT-ATAC (Li et al., 2019) using the distal peaks (non-TSS peaks) of every sample.

As suggested by the reviewer, we have now also inspected the pre-defined JASPAR clusters which our top candidates belong to. Annotated clusters are now presented in **Dataset EV5**. We have observed enrichment of cluster39, which includes IRF1-9, Stat1-2. Of 11 TFs of cluster39 seven were among our top candidates thus confirming the relevance of this cluster of TFs in T-cell biology. Conversely to enrichment of cluster39, we have also identified a small proportion of the members of other clusters to be among our top candidates: 3/94 (3%) TFs of cluster3, 5/56 (9%) TFs of cluster8, 4/32 (13%) TFs of cluster1, and 3/19 (16%) TFs of cluster46. Notably, 2/7 TFs with an unknown function in T-cell biology (HINF and SP4) were the only representatives of their cluster and thus identified by their specific binding features and not by their similarity to other cluster members. (**Dataset EV5**). However, 5/7

TFs with an unknown function in T-cell biology showed binding motifs very similar to other members of the cluster and thus, they may represent spurious hits to known TFs. Taken together, these data indicate that our motif enrichment analysis is specific and not merely driven by the similarities of TF-binding motifs within the different JASPAR clusters.

We have now updated the text of the manuscript to include results of these new in-depth JASPAR cluster analyses (**line 253**):

We have next investigated the hierarchical trees defined by JASPAR (JASPAR clusters), which are based on binding motif similarities. This analysis showed that our top 50 TFs belong to 20 distinct clusters (Dataset EV5). We found that 7/11 (64%) TFs (including IRF1-9 and Stat1-2) that are known to be highly relevant to T-cell development belong to the most represented cluster39 and are among our top candidates. When inspecting the clusters of 7 TFs with no prior known relevance to T-cell biology we noted that the binding motifs of 2 of these (HINF and SP4) do not show similarities to other top candidates (Dataset EV5). HINF is the only TF in cluster76 and SP4 is the only TF among our top 50 belonging to cluster34, which includes a total of 29 TFs, suggesting a previously unrecognized role of HINF and SP4 in T-cell differentiation. By contrast, the binding motifs of 5 of these 7 TFs show similarity with the binding motifs of other members in the list of top 50 (Dataset EV5) and thus, these TFs may merely confirm the role of previously known TFs.

It is surprising that the authors do not attempt to directly compare findings from their analysis of human T-cell ATAC-Seq and RNA-Seq data with corresponding high-quality public datasets obtained in mouse (e.g. by the ImmGen project). Such an analysis could significantly help support some of the authors' claims, since most major regulatory factors would be expected to be conserved in humans and mice, while allowing them to emphasize any findings that are unique to humans, and may therefore reflect the unique value of this data in understanding human T-cell developmental and leukemia. The authors might at least mention the existence of these datasets and suggest that future mouse-human comparisons would be a valuable use of this resource.

Response: As suggested, we have now compared our findings in human cells with previously published work in mice. Specifically, we compared TF binding motif accessibility in human cells to the recently published results of the *cis*-regulatory atlas of the mouse immune system study, conducted in the context of the ImmGen Project (Yoshida et al., 2019). We have now included this in the Discussion section of the manuscript (**line 447**) and updated **Dataset EV5** to indicate TFs which exhibit differential accessibility both, in human and in murine thymic subpopulations.

In the recently published results of the ImmGen Project the expression of murine transcription factors were correlated with changes in chromatin accessibility of TF binding sites. Thereby, a cis-regulatory atlas of 86 primary cell types spanning the mouse immune system was established (Yoshida et al., 2019). Comparison of differential TF binding motifs

in human and mouse datasets showed that 26/50 of our top candidates are also differentially accessible in mouse immune cells, indicating a large degree of conservation of these regulatory factors between human and mice (Dataset EV5). The genome-wide chromatin accessibility atlas of human T-cells that we have generated in this project will thus be a valuable resource for future studies of T-cell maturation and the dysfunction of T-cells in human cancer.

Specific comments:

Introduction: It may be more accurate to say that T-cell leukemias were ‘historically’ subclassified in analogy to T-cell maturation, as the surface marker / maturation stage-based classification mentioned by the authors (Bene et al 1995) is no longer used in clinical diagnostic practice or widely in research. It has generally been superseded by the oncogene-based classification (TAL / LMO / TLX / HOXA etc) plus the ETP-T-ALL maturation category, which overlaps with the earliest stages of the previous classification.

Response: While some clinical studies still use the classification that is based on surface markers we agree with this reviewer that the genetic classification of T-ALL is increasingly being used. As suggested, we have now edited the wording in the Introduction **(line 104)**:

T-cell leukemias are historically subclassified in analogy to T-cell maturation by the expression of the surface markers into four immunophenotypes: pre-, pro-, cortical- and mature- T-ALLs, respectively (Bene et al., 1995).

Suppl figure 1: It is surprising that such a small fraction of ATAC-Seq peaks are in intergenic and intronic regions in all populations. Most distal enhancers are located in these compartments, and they comprise a high fraction of peaks in most high-quality ATAC-Seq datasets from most cell types. It is unclear to me what the authors mean by “non-coding” as a category of genomic region exclusive of intergenic / TSS / TTS / intron / 5’ UTR / 3’ UTR, all of which are themselves non-coding regions. Given that TSS sites were excluded for most of the analysis, it seems that these “non-coding” peaks, as well as TTS sites make up the majority of the peaks analyzed for the differential analysis.

Response: As suggested, we have now more explicitly explained our definition of TSS and non-TSS/distal regions in the main text. We have assigned OCRs to the TSS category if they fell into ± 1 kb of the transcription start site of a CCDS (The Consensus Coding Sequence) gene to improve our classification of the T-cell types. The CCDS project is a collaborative study to identify a core set of human and mouse protein coding regions that are consistently annotated and of high quality (Pruitt et al., 2009), currently comprising approx. 20k genes. All OCRs which did not fall into the ± 1 kb window of a TSS were regarded as “non-TSS/distal” regions. According to this definition, the majority of the peaks (85%) are indeed located in the non TSS/distal areas as already indicated in the text of the manuscript **(line 136)**.

Presumably, this misunderstanding arose from our previous Suppl. Fig. 1 (now removed), which displays categories derived from HOMER analysis (Heinz et al., 2010). These categories

are not directly comparable to those derived from conventional annotations of ATAC-peaks. In the now deleted Suppl. Fig. 1 all categories except the “promoter-TSS” category correspond to the “non-TSS/distal” definition.

As suggested, we have now updated the text of the manuscript (**line 135**) and the Methods (**line 543**) to explain our categorization strategy more explicitly.

Of the 68,415 ATAC-Seq peaks (open chromatin regions; OCRs) (Dataset EV1) identified in the combined analysis of the six donors the majority (85.2%; n = 58,294) fell into distal (non-TSS; outside ± 1 kb window of a TSS) regions while 14.8% (n = 10,121) fell into regions of transcription start sites (TSS; ± 1 kb of a TSS).

OCRs were assigned to the TSS category if they fell into the ± 1 kb window of the transcription start site of a CCDS (The Consensus Coding Sequence, (Pruitt et al., 2009)) gene. All other OCRs, outside the ± 1 kb window, were assigned to the non-TSS/distal category.

The mention of using CIBERSORT to deconvolute T-ALL on lines 161-162 (in the context of figure 2) is confusing, as the actual analysis of T-ALL samples does not occur until a later section (figure 4). It should be clarified that CIBERSORT is being used to validate the signature matrix for normal cells only in this part of the paper.

Response: As suggested, we have now removed this sentence referring to T-ALL deconvolution. We have also updated **line 181** to clarify that CIBERSORT was used for the cross-validation of the signature matrix:

Gradual changes in consecutive developmental stages resulted in poor separation particularly between DN3 and ISP, and double positive stages CD3- and CD3+ (Fig 2A) resulting in a suboptimal prediction accuracy in leave-one-out cross-validations (Appendix Fig S3 A and B) using the CIBERSORT (Newman et al., 2015) deconvolution algorithm to predict cell-types.

Figure 2B: What is the significance of the triangles marked “known regulatory element binding site” (explained in the legend as “known promoter or enhancer regions”)? Through what dataset were these “known enhancers” defined?

Response: This information is retrieved from the “GeneHancer (GH) Regulatory Elements” section of the “GeneCards” database (<https://www.genecards.org/>). GeneHancer is a database of human enhancers and their inferred target genes. It has been developed by integrating a total of 434,000 reported enhancers from four different genome-wide databases: the ENCODE, the Ensembl regulatory build, the FANTOM and the VISTA Enhancer Browser (Fishilevich et al., 2017). In **Fig 2B** triangles indicate the enhancer regions which are annotated in the GeneHancer database and fall into the shown genomic region. We have now updated the figure legend to explain our strategy to define known regions more explicitly:

Fig 2B: ATAC-Seq genome tracks showing three signature peaks located near the genes *SAE1*, *RAG1*, and *CD8A*. Framed regions indicate cell-type-specific signature ATAC-Seq peaks. Blue triangles indicate known promoter or enhancer regions, which are annotated in the GeneHancer (GH) Regulatory Elements database (Fishilevich et al., 2017).

Figure 3a appears to be a schematic diagram explaining the rationale of the ATAC-Seq footprinting analysis rather than actual data, but the sizes depicted are surprising. Distal enhancer NFR peaks detected by ATAC-Seq are typically on the order of 200-300bp, while high-quality TF footprints should be closer to the size of DNA contacted by the TF itself (8-20 bp), though some protected regions are larger due to TF complexes. If the footprints detected in these datasets are really 70 bp in average size, this analysis may just be more accurately defining NFRs, rather than true individual TF footprints. Footprinting analysis with ATAC-Seq has been controversial, as it requires very deeply sequenced and high-quality data, as well as analytic methods that can account for the insertion biases of TN5 transposase (such as the HINT-ATAC method used by the authors). In order for the reader to understand the value of this analysis on this dataset, the authors should show example loci with actual MACS2 / IDR ATAC-Seq peak calls, HINT-ATAC footprinting calls, and example TF motif instances.

Response: As noted by this reviewer, Fig. 3a in the original manuscript reflected a schematic representation of a TF footprinting analysis. As suggested, we have now replaced this panel of Fig. 3a by a direct IGV view of an example locus (*RAG1*) with actual ATAC-Seq peaks with a core size of approx. 230bp, HINT-ATAC footprint calls with an average size of 21bp, and example TF motif instances.

Moreover, we have also analyzed the insert size distribution of ATAC-Seq libraries, which showed the expected ~200bp nucleosome periodicity (**examples below**). Observed peaks at ~200bp, ~400bp, and ~600bp corresponds to one-, two-, and three-nucleosome complexes, which indicates good quality of our sequenced libraries.

They should also show global TF motif enrichment analysis performed on the peak calls alone versus motifs occurring only within footprints, in order to confirm that the footprinting analysis adds value.

Response: As suggested, we have now performed a global motif enrichment analysis only on the ATAC-Seq peaks without intersecting peaks with footprints (**Fig EV3**). Similarly, we have selected the top 50 significant TFs ($p_{adj} < 0.05$), which exhibit the highest standard deviation in the odds ratios to highlight motifs that discriminate between developmental stages. We found that the set of top 50 TFs using peak calls alone contains a higher number of TFs in the category of “unknown role in T-cell development” compared to our motif enrichment analysis using footprints presented in Fig 3B (**20 vs. 7 TFs**). This comparison indicates that we indeed more specifically identify TFs with relevance to T-cell biology by intersecting peaks with footprints compared to using peak calls alone. We have now updated the text of the manuscript accordingly (**line 247**) and have added the new **Fig EV3**:

Furthermore, we have performed a global TF motif enrichment analysis on the ATAC-Seq peaks alone and compared the results of this analysis to enrichment analysis of motifs occurring only within footprints identified in ATAC-peaks. Global analysis yielded a higher number of TFs with no prior known relevance to T-cell biology than intersecting peak calls with footprints (20 vs. 7 TFs; Fig EV3), indicating that footprinting analysis adds substantial specificity towards identifying T-cell related factors.

More detail is required to describe the motif annotation performed with ALFRED and the JASPAR database. Was the analysis restricted to JASPAR CORE vertebrate PFM's? Were

peaks / footprints just overlapped with pre-computed genome-wide motif calls from JASPAR, or were custom thresholds or motif identification criteria used?

Response: As suggested, we have now provided more details of motif calling, peak / footprint annotation and enrichment analysis in the Methods section of the manuscript (**line 601**). Briefly, the motif annotation was indeed restricted to non-redundant JASPAR CORE vertebrate PFMs. We first scanned the GRCh37 reference sequence for genome-wide motif calls using Alfred (Rausch et al., 2018) with default parameters and then intersected these calls with peaks and footprints. Motif enrichment calculation compared counts of a given motif in a given cell type to all other motifs in all other cell types using Fisher's exact test.

The thymus dataset of 58,294 distal peaks was annotated using Alfred (Rausch et al., 2018) with motifs from the non-redundant JASPAR CORE vertebrate PFM catalog (Khan et al., 2018). Only the motifs of TFs which were identified to be expressed in T-cells were considered. To define expressed genes a threshold of >0.5 FPKM was used as defined by the ENCODE project. Motif positions were intersected with footprint positions inferred by HINT-ATAC (Li et al., 2019) using the distal peaks (non-TSS peaks) of every sample.

For figure 3B, the star classification system for literature review requires further explanation. What is the difference between "T-cell development related" and "stage-specific activity" and how are these concepts exclusive of one another?

Response: The "T-cell development related" category includes transcription factors which are generally known to play a role during T-cell differentiation without specifying the stage in which they are active. The "stage-specific" category includes TFs which are known to be active during a specific stage of T-cell differentiation. In this regard all the TFs from the "stage-specific" category are a subset of "T-cell development related" and hence, they are not exclusive. Suppl. Tab. 5 of the original manuscript contains detailed information about the literature review and DOI number of relevant publications. To be more explicit, we have now used more self-explanatory category names in **Fig 3B**, namely **"TFs known to play a role in specific stage of T-cell development"** and **"TFs known to play a role in T-cell development, stage not specified"**.

The authors should discuss the degree to which global differences between T-ALL xenograft and purified normal thymic T cell precursors, such as the greater number of peaks seen in T-ALL samples, might be due to technical issues such as batch effects and / or processing time.

Response: For purified thymic populations we assessed batch effects by collecting thymic samples from six different donors and by preparing and sequencing ATAC-Seq libraries of a subpopulation in different batches (**Dataset EV2**). Thus, we ensured that potential batch effects did not influence the data quality and did not drive the clustering of samples in the PCA (Fig 1C and Fig 5B). For T-ALL patients we assessed batch effects by processing biological replicates of a patient in different batches and showed that different batches give the same signal. We identified a high correlation of 97% (PCC) between the number of reads in ATAC-

Seq peaks of biological replicates (n=10). Moreover, unsupervised clustering of T-ALLs generally clustered biological replicates originating from the same patient together (**below PCA plot**).

Further, we used the same amount of starting material (50,000 cells) while preparing ATAC-Seq libraries of both, purified populations of thymic precursors and T-ALL samples, thus ensuring the same amount of input DNA. The processing time is identical once the 50,000 cells are collected for library preparation. However, processing time until collection of 50,000 purified healthy T-cells was obviously longer than that of leukemia cells because of the FACS sorting step. The probability of cell death is expected to increase with long hours of cell sorting and high pressure using the FACS sorter. Since plasma membranes of dead cells are not intact and the dying process induces morphological changes such as cell shrinkage, cytoplasmic condensation, and nuclear fragmentation (Zhang, Chen, Gueydan, & Han, 2018), the chromatin of dying cells is expected to be disentangled and less organized. However, what we observe was the opposite, namely chromatin of T-ALLs is more open than that of sorted T-cells, indicating that processing time and the FACS sorting procedure prior to cell collection do not introduce artificial chromatin condensation.

In addition, we have now included a table (**Dataset EV2, referred to in line 163 and 328**), which contains the technical aspects of the ATAC-Seq data generation and analysis. We have included the information on the batches (date of the library prep, date of the sequencing, sequencing lane, lot number of the enzyme used, number of PCR cycles, viability of the cells subjected to the ATAC-Seq library preparation and the Ficoll centrifugations details) as well as the quality metrics of the libraries (such as: duplicate fraction, error rate, the number of filtered peaks, fraction of mitochondrial chromosome, peak saturation, fraction mapped to the same chromosome, TSS enrichment, number of unfiltered peaks, unmapped fraction). The quality metrics are well within the range of current standards published by ENCODE. We did not observe that chromatin accessibility profiles are driven by any of the technical parameters presented in **Dataset EV2**. We have now emphasized this in the text of the manuscript (**line 164 and 325**):

Quality control of the libraries revealed that potential batch effects do not detectably influence the data quality and do not drive the clustering of samples in the PCA (Dataset EV2).

Quality control of the libraries revealed that the higher number of OCRs in T-ALLs and the separation of T-ALLs and sorted T-cells are not driven by potential batch effects or other technical parameters detailed in Dataset EV2.

In figure 5C, it is unclear to me what normal progenitor dataset is being used to identify differential gene expression and chromatin accessibility in normal vs. T-ALL - is the comparison to bulk thymus RNA / ATAC data, or to the sorted sub-populations (and if so, which ones)? Both comparisons are mentioned in the methods.

Response: In Fig 5C, as mentioned in the Methods section of the manuscript, we have used sorted populations for the comparison of chromatin accessibility (ATAC-Seq, normal vs. T-ALL) and bulk thymi for the comparison of gene expression (RNA-Seq, normal vs. T-ALL). We could only generate ATAC-Seq libraries and could not perform RNA-Seq from the sorted populations due to restrictions of cell numbers (see above).

We have now updated the legend of **Fig 5C** to explain our differential analyses more explicitly:

Fig 5C: log₂ fold changes (LFC) of 292 genes which are differentially expressed (x-axis, RNA-Seq: non-sorted bulk thymi vs. T-ALLs) and differentially accessible (y-axis, ATAC-Seq: sorted populations vs. T-ALLs) in T-ALLs in comparison to healthy T-cell precursors.

The author's emphasis on DAB1 as a putative novel essential gene in "leukemia" based on having a strong accessibility peak in T-ALL versus normal T-cell progenitors is not well supported by the presented evidence. Despite the finding that DAB1 is upregulated in some T-ALL, the authors note that the absolute quantity of DAB1 transcripts is low (reflected in low FPKM), and it is not clear why the large size of the transcript (which produces a larger absolute number of reads and makes it more readily detectable by RNA-Seq despite a low absolute number of transcripts) is indicative of greater biological significance than its low expression would suggest.

Response: We have now changed the text used in the original version of the manuscript that may have caused misunderstanding. We obviously did not mean to say that the large size of the transcript is indicative of greater biological significance. We wish to point out that the canonical DAB1 transcript that is used for the FPKM calculation is the longest annotated DAB1 transcript whose size exceeds that of the observed DAB1 transcript in our samples by approximately 1kbp. To prevent any further confusion we have now removed the related sentence from the Results section and instead emphasized our new experiments and analyses with regards to the exon usage (**line 363**).

Further, we discovered that T-ALLs express a novel transcript of DAB1 that is not annotated in the GTEx database but expressed in T-ALLs (Fig EV5). Expression of exons 10-15, which are not normally co-expressed, was validated by RT-PCRs and Sanger sequencing in two patients (P7 and P9) and three T-ALL cell lines (Fig EV5). The integration of ATAC-Seq and RNA-Seq data combined with an exon usage analysis thus revealed that the leukemias overexpress a previously unannotated isoform of DAB1 including additional exons. This discovery indicates that the combination of methodologies used here is sensitive to uncover leukemia-specific abnormalities that were not called by previous RNA-Seq analyses alone. Future work will be required to investigate whether this previously unrecognized transcript of DAB1 affects the viability or proliferation of leukemia cells in T-ALL.

The authors reference public DepMap RNAi screen data performed on 25 leukemia cell lines, only 2 of which are T-ALL. The average DEMETER dependency score of -0.21 across “leukemia” is not an especially strong effect. Unmentioned by the authors is the fact the two T-ALL cell lines show among the weakest effects of any leukemia cell line. (both > -0.1, likely insignificant). Furthermore, more specific CRISPR screen data is also available through DepMap, and shows DAB1 dependency scores near zero for three T-ALL cell lines, and a near-zero average score for leukemia cell lines in general, indistinguishable from other cancer types (with the possible exception of peripheral nerve tumors). In the absence of other experimental support for a functional role of DAB1 in T-ALL, this emphasis should be substantially reduced.

Response: : We have used the Combined RNAi (Broad, Novartis, Marcotte) dataset of DepMap portal since it contained information for 25 leukemia cell lines, two of which are T-ALL cell lines with which we used for qPCR experiments. This was the closest dataset that is likely to represent T-cell leukemia in the context of RNAi screen data. Nevertheless, as the reviewer pointed out, there is no especially strong effect for DAB1 and it remains inconclusive from this analysis what the real effects of *DAB1* are in T-ALL.

In new experiments, we have now investigated differential exon usage in the RNA-Seq data of T-ALL patients and bulk thymi and found that a previously unannotated RNA transcript which is not annotated in the GTEx portal is expressed in T-ALL patients but not in normal thymus. According to GTEx annotations there is only one isoform of DAB1 containing exon 10. In this isoform exon 10 is co-expressed with exons 11 and 12 but not with exons 2, 3, 10-15 and 26. Our differential exon usage analysis shows an atypical isoform containing exon 10 together with exons 2, 3, 10-15 and 26 (Fig EV5). We have now targeted a region spanning exons 10-15 by RT-PCR and Sanger sequencing, and validated its presence in three T-ALL cell lines and in two patients (P7 and P9) of whom RNA of sufficient quantity has been available (Fig EV5). As outlined above, the integration of ATAC-Seq and RNA-Seq data can improve the power to uncover leukemia-specific abnormalities that were not called previously by RNA-Seq analyses alone. We now show these data in the new Fig EV5, referred to in line 363.

Furthermore, DepMap RNAi screens have targeted four of the *DAB1* exons (exons 12, 13, 17 and 22), only two of which are found to be expressed in T-ALL. Due to underrepresentation of T-ALL in the DepMap portal and our findings identifying a novel *DAB1* transcript in T-ALL, whose majority (7/9) of the exons have not been targeted by RNAi screens, we have decided to replace the DepMap dependency score analysis with the more informative results of the differential exon usage analysis (line 365 and Fig EV5):

Further, we discovered that T-ALLs express a novel transcript of DAB1 that is not annotated in the GTEx database but expressed in T-ALLs (Fig EV5). Expression of exons 10-15, which are not normally co-expressed, was validated by RT-PCRs and Sanger sequencing in two patients (P7 and P9) and three T-ALL cell lines (Fig EV5). The integration of ATAC-Seq and RNA-Seq data combined with an exon usage analysis thus revealed that the leukemias overexpress a previously unannotated isoform of DAB1 including additional exons. This discovery indicates that the combination of methodologies used here is sensitive to uncover leukemia-specific abnormalities that were not called by previous RNA-Seq analyses alone. Future work will be required to investigate whether this previously unrecognized transcript of DAB1 affects the viability or proliferation of leukemia cells in T-ALL.

As suggested, we have now reduced the emphasis on *DAB1* by updating the title in the Results section of the manuscript as well (line 329):

T-ALLs form two subgroups characterized by overexpression of *DAB1* or *SPI1*

Referee #2 (Remarks for Author):

This is an interesting study of T-ALL and normal T-cell development based on ATAC-sequencing, transcription factor motif analyses and gene expression data. Overall, the authors reveal the open/closed chromatin regions in T-cell subsets and T-ALL samples by ATAC-seq and use that information to compare T-ALL with T-cell subsets. They discover that T-ALL cells are most similar to immature T-cells in DN2/DN3 stage. Based on transcription factor motifs present in the accessible regions, they estimate which transcription factors are most likely implicated in the regulation of gene expression. Moreover, in additional analyses with gene expression data, the authors identify *DAB1* as a gene that is upregulated in T-ALL.

This work is of interest and provides novel views on the regulation of T-cell and T-ALL development, but remains largely descriptive as no genes are inactivated or overexpressed to study the observations made in more detail (except for the use of publicly available loss-of-function screen data for *DAB1*).

Major comments:

- The abstract is misleading and seems to indicate that the authors study the cell of origin of T-ALL, which is not done at all. Moreover, the emphasis on *SUMO1* and *DAB1* is overestimating the potential role that these factors play in T-ALL as there is no proof for their importance (only expression is documented).

Response: We agree with this reviewer that the term “Cell of origin” may be misleading. We used the term for cells at the developmental stage with the highest resemblance to T-ALLs in terms of chromatin accessibility. We have now rephrased the abstract (detailed below), the running title (**line 67**), the introduction (**line 118**), the results (**line 391**) and the discussion (**line 406 and 498**) accordingly:

ATAC-Seq of normal and leukemia T-cells

To identify the maturation stages closest to T-ALLs and thus where T-ALL cells are likely arrested we employed the Assay for Transposase Accessible Chromatin Sequencing (ATAC-Seq)...

... and place the T-cell developmental stage, pediatric T-ALLs likely originate, at different albeit consistently early levels of the hierarchy of the developmental pathway of maturing T-cells.

... and defining the thymic maturation stage leukemia cells likely originate ...

... and identify particularly immature developmental stages to be the closest to pediatric T-ALLs, which are characterized either by activated *DAB1* or *SPI1* in our patient cohort.

In new experiments we have now taken additional steps to validate the role of *DAB1*, which in the original manuscript we reported to be most recurrently over-expressed and hyper-accessible in T-ALL patients and to carry the most discriminating capacity between normal and leukemic T-cells. We have now analyzed differential exon usage in the RNA-Seq data of T-ALL patients and bulk thymi and found that a previously unannotated RNA transcript which is not annotated in the GTEx portal is expressed in T-ALL patients but not in normal thymus. According to GTEx annotations there is only one isoform of *DAB1* containing exon 10. In this isoform exon 10 is always co-expressed with exons 11 and 12 but not with exons 2, 3, 10-15 and 26. In our analysis of differential exon usage we observe an atypical isoform containing exon 10 together with exons 2, 3, 10-15 and 26. (**Fig EV5**). We have now targeted a region spanning exons 10-15 by RT-PCR and **Sanger sequencing** and validated its presence in three T-ALL cell lines and in two patients (P7 and P9) of whom RNA of sufficient quantity has been available (**Fig EV4**). As outlined above, the integration of ATAC-Seq and RNA-Seq data can improve the power to uncover leukemia-specific abnormalities that were not called previously by RNA-Seq analyses alone. We now show these data in the new **Fig EV5**, referred to in **line 363**:

Further, we discovered that T-ALLs express a novel transcript of *DAB1* that is not annotated in the GTEx database but expressed in T-ALLs (Fig EV5). Expression of exons 10-15, which are not normally co-expressed, was validated by RT-PCRs and Sanger sequencing in two patients (P7 and P9) and three T-ALL cell lines (Fig EV5). The integration of ATAC-Seq and RNA-Seq data combined with an exon usage analysis thus revealed that the leukemias overexpress a previously unannotated isoform of *DAB1* including additional

exons. This discovery indicates that the combination of methodologies used here is sensitive to uncover leukemia-specific abnormalities that were not called by previous RNA-Seq analyses alone. Future work will be required to investigate whether this previously unrecognized transcript of *DAB1* affects the viability or proliferation of leukemia cells in T-ALL.

As suggested, we have toned down the emphasis on *SAE1* (SUMO1 Activating Enzyme) and SUMOylation in T-cell development in the abstract (line 70):

We aimed at identifying the developmental stage at which leukemic cells of pediatric T-ALLs are arrested and at defining leukemogenic mechanisms based on ATAC-Seq. Chromatin accessibility maps of seven developmental stages of human healthy T-cells revealed progressive chromatin condensation during T-cell maturation. Developmental stages were distinguished by 2,823 signature chromatin regions with 95% accuracy. Open chromatin surrounding *SAE1* was identified to best distinguish thymic developmental stages suggesting a potential role of SUMOylation in T-cell development. Deconvolution using signature regions revealed that T-ALLs, including those with mature immunophenotypes, resemble the most immature populations, which was confirmed by TF binding-motif profiles. We integrated ATAC-Seq and RNA-Seq and found *DAB1*, a gene not related to leukemia previously, to be overexpressed, abnormally spliced and hyper-accessible in T-ALLs. *DAB1*-negative patients formed a distinct sub-group with particularly immature chromatin profile and hyper-accessible binding sites for *SPI1* (*PU.1*), a TF crucial for normal T-cell maturation. In conclusion, our analyses of chromatin accessibility and TF binding-motifs showed that pediatric T-ALL cells are most similar to immature thymic precursors, indicating an early developmental arrest.

- Figure 1: rather than to only show the total number of ATAC-seq peaks in figure 1, and to show that the total number is decreasing upon differentiation of T-cells, it would be also informative to show the number of regions that become less accessible and also the number of regions that become more accessible - if there are such regions.

Response: As suggested, we have now divided 68,415 OCRs into four categories according to the changes in accessibility patterns throughout differentiation: increasing (stdev in read counts (RCs) ≥ 12 and RCs of consecutive stages increase), decreasing (stdev ≥ 12 and RCs of consecutive stages decrease), fluctuating (stdev ≥ 12 and RCs going up/down during development) and steady (stdev ≤ 12 or stdev ≥ 12 and difference between RCs of consecutive stages between -24 and 24) (Fig 1C). This analysis showed that the accessibility pattern of the majority of OCRs (58.7%) remains steady during thymocyte maturation. Moreover, steady peaks tended to have less accessibility (mean peak count: 20), indicating that closed/less accessible chromatin regions in early development tend to remain closed as T-cells mature. We found that only 0.3% of OCRs become more accessible, whereas 28.7% become less accessible during thymocyte maturation. These findings suggest that changes of chromatin accessibility occurring in the course of T-cell development are in the direction of

“closing/condensing” the accessible gene regulatory elements instead of “opening/decondensing” the packed gene regulatory elements, which corroborates our findings shown in **Fig 1B**. We now describe these new results in a new **panel C of Figure 1** and in the text of the Results section (**line 151**):

Moreover, we have categorized 68,415 OCRs into four patterns (increasing, decreasing, fluctuating and steady) based on the changes in accessibility patterns and found that the majority of OCRs (59%) remain steady during thymocyte maturation (Fig 1C). Steady peaks tended to show less accessibility (mean peak count: 20), indicating that less open chromatin regions in early development tend to remain closed as T-cells develop. We found that 29% of OCRs became less accessible, whereas only 0.3% became more accessible demonstrating that chromatin organization in developing thymocytes is characterized by closing/condensing those regions that are highly accessible in the immature precursors (Fig 1C).

We also discussed these results in the Discussion section (**line 424**):

The higher fraction of OCRs with a closing profile than those with an opening profile (29% vs. 0.3%) in the course of maturation implicates that chromatin remodeling during T-cell maturation is mediated by condensing accessible regions instead of decondensing packed regions.

We have also updated the Methods section accordingly (**line 547**):

Peak categorization into four patterns (increasing, decreasing, fluctuating and steady) was based on the standard deviation in the normalized read counts and the accessibility profile of a peak in the course of maturation. Increasing: $stdev \geq 12$ and RCs of consecutive stages increase, decreasing: $stdev \geq 12$ and RCs of consecutive stages decrease, fluctuating: $stdev \geq 12$ and RCs going up/down during development, steady: $stdev \leq 12$ or $stdev \geq 12$ and difference between RCs of consecutive stages between -24 and 24.

- For the transcription factors shown in figure 3: are these transcription factors expressed at these stages of T-cell development? Such expression data should be taken into account if possible.

Response: As outlined in detail in the response to reviewer #1, the numbers of cells in the rare fractions of the thymic subpopulations did not suffice to perform RNA-Seq analyses. We have therefore mined existing data sets for information about expression and/or function of the transcription factors in T-cell development. **Dataset EV5** of the original manuscript contains detailed information about the literature review and DOI number of relevant publications. The star classification system in **Fig 3B** of the original manuscript indicates three categories of TFs. We now use a more self-explanatory nomenclature of the different categories **“TFs known to play a role in specific stage of T-cell development”** and **“TFs known to play a role in T-cell development, stage not specified”**.

- If the authors hypothesize that SUMO1 is important: is there also publicly available loss-of-function screen data for this gene (as used for DAB1)? This is currently all hypothetical.

Response: *SAE1* (SUMO1 Activating Enzyme) carries the highest capacity to distinguish purified thymic populations suggesting that this enzyme and tentatively SUMOylation may play a role in normal T-cell differentiation. As suggested, we have mined publicly available data in more detail and have introduced the following new sentences into the Discussion section of the manuscript (**line 434**):

Moreover, *SAE1* is required for cell-autonomous definitive hematopoiesis, suggesting *SAE1* importance for maintenance of hematopoietic stem/progenitor (Li, Lan, Xu, Zhang, & Wen, 2012). In this context, it is interesting to note that the analysis of chromatin accessibility reported here suggests a potential role of *SAE1* in normal T-cell development.

- For DAB1: the authors use information from a publicly available loss-of-function screen. Can a more broad analysis be performed where the ATAC-seq/gene expression data is compared with the loss-of-function screen for all the top candidate genes that are similar to DAB1 (specifically expressed in T-ALL and in accessible regions) ?

Response: Publicly available LoF screen data on the Cancer Dependency Map portal were generated from 25 leukemia cell lines, only two of which are T-ALLs. Moreover, our new analysis of differential exon usage showed that the *DAB1* transcript that is expressed in T-ALL has not actually been annotated before. DepMap RNAi screens have targeted four of the *DAB1* exons (exons 12, 13, 17 and 22), only two of which are found to be expressed in T-ALL. Due to underrepresentation of T-ALL in the DepMap portal and our findings identifying a novel *DAB1* transcript in T-ALL, whose majority (7/9) of the exons have not been targeted by RNAi screens, we have decided to eliminate the dependency scores as a line of argumentation. These analyses are now replaced by differential exon usage analyses (**line 363 and Fig EV5**), whose details can be found in response to referee 1.

Minor comments:

- The authors state “Chromatin accessibility as measured by the number of OCRs decreased progressively with maturation, particularly at promoter and non-coding regions (Suppl. Fig. 1).” Can the authors document this more detailed towards promoter and known transcriptional regulatory regions (for example H3K27ac regions)? Also, what exactly are the regions defined in Suppl fig 1 (for example ‘transcription termination site’ - how are these regions defined and how large are these regions ? 1 kb around termination site or what exactly ?); so it remains poorly specified where the OCRs decrease exactly.

Response: As suggested, we have now compared open chromatin regions identified in thymic developmental stages with previously published ChIP-Seq data for the T-ALL cell line DND-41. This analysis showed an enrichment of active chromatin marks such as H3K27ac and H3K4me3 in OCRs, whereas repressed chromatin marks such as H3K9me3 are depleted

in OCRs that we identified. We have now summarized these analyses and results in **Fig EV1** and the text of the manuscript (**line 169**):

A comparison with previously published methylation/acetylation datasets for the T-ALL cell line DND-41 (Knoechel et al., 2014), in which we computed expected values based on randomly shuffled ATAC-Seq peaks, shows a high degree of overlap between the OCRs identified in purified subpopulations and the active promoters and enhancers detected in CHIP-Seq datasets (Fig EV1).

The categories shown in Suppl. Fig. 1 of the original manuscript were annotated using HOMER software (Heinz et al., 2010). “Transcription termination site” region is defined from -100bp to +1kb of TTS and “transcription start site region” is defined from -1kb to +100bp of a TSS by HOMER. However, by convention, we have divided OCRs into two categories: TSS and non-TSS/distal. As discussed in detail in response to referee 1, we have assigned OCRs to the TSS category if they fall into ± 1 kb of transcription start site of a CCDS (The Consensus Coding Sequence) gene to improve our classification of the T-cell types. All OCRs outside this ± 1 kb window of a TSS were regarded as “non-TSS/distal” regions. Thanks to comments of both reviewers we identified the inconsistencies between our and HOMER’s definition of TSSs and therefore decided to remove Suppl. Fig. 1. We have now updated the text of the manuscript (**line 135**) and the Methods (**line 543**) to explain categorization strategy more explicitly.

Of the 68,415 ATAC-Seq peaks (open chromatin regions; OCRs) (Dataset EV1) identified in the combined analysis of the six donors the majority (85.2%; n = 58,294) fell into distal (non-TSS; outside ± 1 kb window of a TSS) regions while 14.8% (n = 10,121) fell into regions of transcription start sites (TSS; ± 1 kb of a TSS).

OCRs were assigned to the TSS category if they fell into the ± 1 kb window of the transcription start site of a CCDS (The Consensus Coding Sequence, (Pruitt et al., 2009)) gene. All other OCRs, outside the ± 1 kb window, were assigned to the non-TSS/distal category.

- The authors find that the expression of DAB1 is 41-times higher in the T-ALLs than in bulk thymus. It would be more appropriate to check DAB1 expression in the different normal subsets of the thymus, since each of the subsets may show different DAB1 levels - bulk thymus expression level is not very informative, especially since the authors show that T-ALL is more comparable to DN2/DN3 stages than to the more differentiated stages or to the DP stage which is the most frequent cell type in thymus.

Response: As discussed above we had to use the sorted T-cells to generate ATAC-Seq libraries. The numbers of sorted cells did not suffice to generate RNA-Seq libraries. However, to address this point, we have analyzed publicly available microarray expression profiling data of sorted human thymocyte populations (GEO Accession number: GSE33470) with GEO2R. Differential expression analysis showed that *DAB1* expression is not significantly

different between sorted populations (padj=0.39; **Appendix Fig S8**). *DAB1* expression is as stable as *GAPDH* (padj=0.48; **Appendix Fig S8**). We have also observed that *DAB1* expression in normal thymic precursors is even lower than *HBA1* (Hemoglobin Subunit Alpha 1, padj=0.84), which is known not to be expressed in T-cells. As a control we have also inspected genes which were expected to be significantly differentially expressed in certain developmental stages such as *CD4* (padj=0.0063) and *RAG1* (padj=0.0001). We have now updated the text of the manuscript to include the results of the published microarray analysis (**line 352 and Appendix Fig S8**):

We have also analyzed publicly available microarray expression profiling data of sorted human thymocyte populations (GEO Accession No.: GSE33470) and showed that *DAB1* expression is exceedingly low and similar to the expression of *HBA1* (Hemoglobin Subunit Alpha 1, which is known not to be expressed in T-cells in all subpopulations (Appendix Fig S8). These data indicate that RNA-Seq of bulk thymi is representative for the expression of *DAB1* in all subpopulations.

We have also updated the Methods section accordingly (**line 675**):

Differential expression analysis of publicly available microarray dataset of sorted human thymocyte populations was performed with GEO2R (Barrett et al., 2013).

References

- Arber, D. A., Orazi, A., Hasserjian, R., Thiele, J., Borowitz, M. J., Le Beau, M. M., . . . Vardiman, J. W. (2016). The 2016 revision to the World Health Organization classification of myeloid neoplasms and acute leukemia. *Blood*, *127*(20), 2391-2405. Retrieved from <https://www.ncbi.nlm.nih.gov/pubmed/27069254>. doi:10.1182/blood-2016-03-643544
- Corces, M. R., Buenrostro, J. D., Wu, B., Greenside, P. G., Chan, S. M., Koenig, J. L., . . . Chang, H. Y. (2016). Lineage-specific and single-cell chromatin accessibility charts human hematopoiesis and leukemia evolution. *Nat Genet*, *48*(10), 1193-1203. Retrieved from <https://www.ncbi.nlm.nih.gov/pubmed/27526324>. doi:10.1038/ng.3646
- Fishilevich, S., Nudel, R., Rappaport, N., Hadar, R., Plaschkes, I., Iny Stein, T., . . . Cohen, D. (2017). GeneHancer: genome-wide integration of enhancers and target genes in GeneCards. *Database (Oxford)*, *2017*. Retrieved from <https://www.ncbi.nlm.nih.gov/pubmed/28605766>. doi:10.1093/database/bax028
- Heinz, S., Benner, C., Spann, N., Bertolino, E., Lin, Y. C., Laslo, P., . . . Glass, C. K. (2010). Simple combinations of lineage-determining transcription factors prime cis-regulatory elements required for macrophage and B cell identities. *Mol Cell*, *38*(4), 576-589. Retrieved from <https://www.ncbi.nlm.nih.gov/pubmed/20513432>. doi:10.1016/j.molcel.2010.05.004
- Kessler, J. D., Kahle, K. T., Sun, T., Meerbrey, K. L., Schlabach, M. R., Schmitt, E. M., . . . Westbrook, T. F. (2012). A SUMOylation-dependent transcriptional subprogram is required for Myc-driven tumorigenesis. *Science*, *335*(6066), 348-353. Retrieved from <https://www.ncbi.nlm.nih.gov/pubmed/22157079>. doi:10.1126/science.1212728
- Khan, A., Fornes, O., Stigliani, A., Gheorghe, M., Castro-Mondragon, J. A., van der Lee, R., . . . Mathelier, A. (2018). JASPAR 2018: update of the open-access database of transcription factor binding profiles and its web framework. *Nucleic Acids Res*, *46*(D1), D260-D266. Retrieved from <https://www.ncbi.nlm.nih.gov/pubmed/29140473>. doi:10.1093/nar/gkx1126
- Knoechel, B., Roderick, J. E., Williamson, K. E., Zhu, J., Lohr, J. G., Cotton, M. J., . . . Bernstein, B. E. (2014). An epigenetic mechanism of resistance to targeted therapy in T cell acute lymphoblastic leukemia. *Nat Genet*, *46*(4), 364-370. Retrieved from <http://www.ncbi.nlm.nih.gov/pubmed/24584072>. doi:10.1038/ng.2913
- Li, X., Lan, Y., Xu, J., Zhang, W., & Wen, Z. (2012). SUMO1-activating enzyme subunit 1 is essential for the survival of hematopoietic stem/progenitor cells in zebrafish. *Development*, *139*(23), 4321-4329. Retrieved from <https://www.ncbi.nlm.nih.gov/pubmed/23132242>. doi:10.1242/dev.081869
- Liu, Y., Easton, J., Shao, Y., Maciaszek, J., Wang, Z., Wilkinson, M. R., . . . Mullighan, C. G. (2017). The genomic landscape of pediatric and young adult T-lineage acute lymphoblastic leukemia. *Nat Genet*, *49*(8), 1211-1218. Retrieved from <https://www.ncbi.nlm.nih.gov/pubmed/28671688>. doi:10.1038/ng.3909
- Meijerink, J. P. P. (2010). Genetic rearrangements in relation to immunophenotype and outcome in T-cell acute lymphoblastic leukaemia. *Best Practice & Research Clinical Haematology*, *23*(3), 307-318. Retrieved from <Go to ISI>://WOS:000285663900002. doi:10.1016/j.beha.2010.08.002
- Newman, A. M., Liu, C. L., Green, M. R., Gentles, A. J., Feng, W., Xu, Y., . . . Alizadeh, A. A. (2015). Robust enumeration of cell subsets from tissue expression profiles. *Nat*

- Methods*, 12(5), 453-457. Retrieved from <https://www.ncbi.nlm.nih.gov/pubmed/25822800>. doi:10.1038/nmeth.3337
- Palomero, T., Lim, W. K., Odom, D. T., Sulis, M. L., Real, P. J., Margolin, A., . . . Ferrando, A. A. (2006). NOTCH1 directly regulates c-MYC and activates a feed-forward-loop transcriptional network promoting leukemic cell growth. *Proc Natl Acad Sci U S A*, 103(48), 18261-18266. Retrieved from <https://www.ncbi.nlm.nih.gov/pubmed/17114293>. doi:10.1073/pnas.0606108103
- Pruitt, K. D., Harrow, J., Harte, R. A., Wallin, C., Diekhans, M., Maglott, D. R., . . . Lipman, D. (2009). The consensus coding sequence (CCDS) project: Identifying a common protein-coding gene set for the human and mouse genomes. *Genome Res*, 19(7), 1316-1323. Retrieved from <https://www.ncbi.nlm.nih.gov/pubmed/19498102>. doi:10.1101/gr.080531.108
- Rausch, T., Fritz, M. H., Korb, J. O., & Benes, V. (2018). Alfred: Interactive multi-sample BAM alignment statistics, feature counting and feature annotation for long- and short-read sequencing. *Bioinformatics*. Retrieved from <https://www.ncbi.nlm.nih.gov/pubmed/30520945>. doi:10.1093/bioinformatics/bty1007
- Richter-Pechanska, P., Kunz, J. B., Bornhauser, B., von Knebel Doeberitz, C., Rausch, T., Erarslan-Uysal, B., . . . Kulozik, A. E. (2018). PDX models recapitulate the genetic and epigenetic landscape of pediatric T-cell leukemia. *EMBO Mol Med*, 10(12). Retrieved from <https://www.ncbi.nlm.nih.gov/pubmed/30389682>. doi:10.15252/emmm.201809443
- Weng, A. P., Millholland, J. M., Yashiro-Ohtani, Y., Arcangeli, M. L., Lau, A., Wai, C., . . . Aster, J. C. (2006). c-Myc is an important direct target of Notch1 in T-cell acute lymphoblastic leukemia/lymphoma. *Genes Dev*, 20(15), 2096-2109. Retrieved from <https://www.ncbi.nlm.nih.gov/pubmed/16847353>. doi:10.1101/gad.1450406
- Yan, F., Powell, D. R., Curtis, D. J., & Wong, N. C. (2020). From reads to insight: a hitchhiker's guide to ATAC-seq data analysis. *Genome Biol*, 21(1), 22. Retrieved from <https://www.ncbi.nlm.nih.gov/pubmed/32014034>. doi:10.1186/s13059-020-1929-3
- Yoshida, H., Lareau, C. A., Ramirez, R. N., Rose, S. A., Maier, B., Wroblewska, A., . . . Immunological Genome, P. (2019). The cis-Regulatory Atlas of the Mouse Immune System. *Cell*, 176(4), 897-912 e820. Retrieved from <https://www.ncbi.nlm.nih.gov/pubmed/30686579>. doi:10.1016/j.cell.2018.12.036
- Zhang, Y., Chen, X., Gueydan, C., & Han, J. (2018). Plasma membrane changes during programmed cell deaths. *Cell Res*, 28(1), 9-21. Retrieved from <https://www.ncbi.nlm.nih.gov/pubmed/29076500>. doi:10.1038/cr.2017.133
- Barrett, T., Wilhite, S. E., Ledoux, P., Evangelista, C., Kim, I. F., Tomashevsky, M., . . . Soboleva, A. (2013). NCBI GEO: archive for functional genomics data sets--update. *Nucleic Acids Res*, 41(Database issue), D991-995. Retrieved from <https://www.ncbi.nlm.nih.gov/pubmed/23193258>. doi:10.1093/nar/gks1193
- Khan, A., Fornes, O., Stigliani, A., Gheorghe, M., Castro-Mondragon, J. A., van der Lee, R., . . . Mathelier, A. (2018). JASPAR 2018: update of the open-access database of transcription factor binding profiles and its web framework. *Nucleic Acids Res*, 46(D1), D260-D266. Retrieved from <https://www.ncbi.nlm.nih.gov/pubmed/29140473>. doi:10.1093/nar/gkx1126

- Knoechel, B., Roderick, J. E., Williamson, K. E., Zhu, J., Lohr, J. G., Cotton, M. J., . . . Bernstein, B. E. (2014). An epigenetic mechanism of resistance to targeted therapy in T cell acute lymphoblastic leukemia. *Nat Genet*, *46*(4), 364-370. Retrieved from <http://www.ncbi.nlm.nih.gov/pubmed/24584072>. doi:10.1038/ng.2913
- Li, Z., Schulz, M. H., Look, T., Begemann, M., Zenke, M., & Costa, I. G. (2019). Identification of transcription factor binding sites using ATAC-seq. *Genome Biol*, *20*(1), 45. Retrieved from <https://www.ncbi.nlm.nih.gov/pubmed/30808370>. doi:10.1186/s13059-019-1642-2
- Pruitt, K. D., Harrow, J., Harte, R. A., Wallin, C., Diekhans, M., Maglott, D. R., . . . Lipman, D. (2009). The consensus coding sequence (CCDS) project: Identifying a common protein-coding gene set for the human and mouse genomes. *Genome Res*, *19*(7), 1316-1323. Retrieved from <https://www.ncbi.nlm.nih.gov/pubmed/19498102>. doi:10.1101/gr.080531.108
- Rausch, T., Fritz, M. H., Korb, J. O., & Benes, V. (2018). Alfred: Interactive multi-sample BAM alignment statistics, feature counting and feature annotation for long- and short-read sequencing. *Bioinformatics*. Retrieved from <https://www.ncbi.nlm.nih.gov/pubmed/30520945>. doi:10.1093/bioinformatics/bty1007

Thank you for the submission of your revised manuscript to EMBO Molecular Medicine. We have now received the enclosed report from the two referees who were asked to re-assess it.

You will see from the comments below that while referee #2 is overall satisfied with the revision, referee #1 indicated that due to the elimination of the claim for a functional role of DAB1, the revised manuscript does not seem to provide validated specific findings of clear medical or biological importance. As such, this referee indicated that she/he does not support publication of the manuscript in its current form.

In principle, our editorial policy only allows a single round of major revision. However, as indicated in the previous decision letter, we think that it is essential to experimentally address referee #1's concern in this regard and to provide certain levels of insights into the functional role of DAB1 in the revised manuscript, to enhance the overall medical impact of the study. We would therefore ask you to address this point in an exceptional second round of revision.

***** Reviewer's comments *****

Referee #1 (Remarks for Author):

The authors have appropriately addressed the concerns raised in my original review, although the revisions have eliminated significant claims made in the original version with regard to the functional importance of DAB1, a "punchline" finding in the original. In place of the claims for an essential functional role of DAB1, the authors claim to have described a novel DAB1 splice form, but this finding is not very completely described (see below). This work presents what appears to be a highly valuable data resource for the T-ALL research community, but without validated specific findings of clear medical or biological importance.

Referee #2 (Remarks for Author):

The authors have responded well to the comments and suggestions. The manuscript is now improved and I have no other questions.

Thank you for your message regarding our recent decision on your manuscript. I have considered the points raised in your letter and have also passed down the message to referee #1. Referee #1 is now supportive of publication of the revised manuscript and I am pleased to inform you that we will be able to accept your manuscript pending the following essential amendments.

2nd Authors' Response to Reviewers**19th Jun 2020**

The Authors have made the requested editorial changes.

Please find enclosed the final reports on your manuscript. We are pleased to inform you that your manuscript is accepted for publication and is now being sent to our publisher to be included in the next available issue of EMBO Molecular Medicine.

Corresponding Author Name: Andreas Kulozik and Jan Korbelt

Manuscript Number: EMM-2020-12104